# Imitation Learning with Temporal Logic Constraints

**Zining Fan**
Rutgers University
zf140@scarletmail.rutgers.edu

**He Zhu**
Rutgers University
hz375@cs.rutgers.edu

## Abstract

Designing reinforcement learning agents to satisfy complex temporal objectives expressed in Linear Temporal Logic (LTL), presents significant challenges, particularly in ensuring sample efficiency and task alignment over infinite horizons. Recent works have shown that by leveraging the corresponding Limit Deterministic Büchi Automaton (LDBA) representation, LTL formulas can be translated into variable discounting schemes over LDBA-accepting states to maximize a lower bound on the probability of formula satisfaction. However, the resulting reward signals are inherently sparse, making exploration of LDBA-accepting states increasingly difficult as task horizons lengthen to infinity. In this work, we address these challenges by leveraging finite-length demonstrations to overcome the exploration bottleneck for LTL objectives over infinite horizons. We segment agent exploratory trajectories at LDBA-accepting states and iteratively guide the agent within each segment to learn to efficiently reach these accepting states. By incentivizing the agent to visit LDBA-accepting states from arbitrary states, our approach increases the probability of LTL formula satisfaction without the need for extensive or lengthy demonstrations. We demonstrate the applicability of our method in a variety of high-dimensional continuous control domains. It achieves faster convergence and consistently outperforms baseline approaches.

## 1 Introduction

Linear Temporal Logic (LTL) has been extensively studied as an alternative framework for specifying objectives for reinforcement learning (RL) agents [54, 27, 13, 62, 3, 16, 63]. LTL provides a powerful and flexible language to define tasks with temporal dependencies, such as "cycle between two subgoals while always avoiding unsafe regions" or "eventually reach a goal after completing a sequence of subtasks" [2]. Designing RL agents to satisfy these objectives is particularly challenging when considering infinite horizons, where the agent must maintain behavior that satisfies the objectives indefinitely. These challenges are compounded by the need to ensure sample efficiency in high-dimensional, continuous systems.

Several recent studies [28, 63, 13] have proposed proxy reward schemes to derive policies from the Limit Deterministic Büchi Automaton (LDBA) representation of LTL specifications. A trajectory satisfies an LTL formula if and only if it visits an LDBA-accepting state infinitely often. However, these proxy rewards, defined over LDBA-accepting states, are inherently sparse, posing challenges for effective exploration toward such states.

To address these limitations, we propose a novel framework, TiLoIL (Temporal Logic Imitation Learning), which leverages Learning from Demonstrations to mitigate reward sparsity in policy optimization for LTL objectives. The high-level idea is to combine the structure of LDBAs with finite-length demonstrations to guide exploration. Specifically, TiLoIL segments exploratory agent trajectories at each visit to an LDBA accepting state, treating each segment as a sub-trajectory leading toward acceptance. Using these segments, TiLoIL encourages the agent to learn how to efficiently reach LDBA accepting states by imitating expert behavior along these sub-trajectories. In essence,

39th Conference on Neural Information Processing Systems (NeurIPS 2025).

TiLoIL encourages the agent to consistently seek LDBA-accepting states from any non-accepting states, thereby increasing the likelihood of satisfying infinite-horizon LTL formulas. Along the way, TiLoIL learns a reward function for reaching LDBA-accepting states by contrasting successful trajectories with unsuccessful ones. This reward function is then leveraged for policy training, hence reducing the need for extensive or lengthy demonstration data.

Moreover, TiLoIL uses the inherent multistage structure of LTL formulas to further densify the reward function to reach the LDBA-accepting states. At each stage, the reward function provides the agent with rich reward signals specifically tailored to that stage. This staged approach improves the efficiency of the learning process compared to treating LDBA-accepting state reaching as a single monolithic procedure.

We demonstrate the effectiveness of TiLoIL across a range of tasks in high-dimensional continuous systems, which have historically posed significant challenges for LTL-based RL methods. Our method achieves faster convergence and outperforms baseline approaches, demonstrating superior generalization to unseen scenarios. By leveraging the synergy between LTL specifications, LDBA representations, and demonstration data, TiLoIL provides a practical solution to design RL agents that satisfy complex temporal objectives.

## 2 Background and Problem Setup

This section sets up our reinforcement learning problem to solve tasks specified by Linear Temporal Logic (LTL).

### 2.1 Linear Temporal Logic (LTL)

LTL [50] is a specification language that combines atomic propositions ($AP$s) and logical operators to describe system behaviors and temporal properties. An atomic proposition $AP$ represents a basic indivisible statement about the state of a system that can be true or false at a given time. The logical operators include: not ($\neg$), and ($\wedge$), and implies ($\rightarrow$); and the temporal operators include: next ($\mathsf{X}$), repeatedly/always/globally ($\mathsf{G}$), eventually ($\mathsf{F}$), and until ($\mathsf{U}$). Appendix C defines the syntax and semantics of LTL formulas. For a complete introduction, we refer the reader to Baier and Katoen [7].

**Example.** Consider the *FlatWorld* environment in Fig. 1 with $AP = \{y, g, r, b\}$ where $r$ labels the red region, $g$ labels the green, $y$ labels the yellow, and $b$ labels the blue. If the task is to eventually reach the yellow zone and remain there, we express this as $\varphi = \mathsf{FG}y$. Alternatively, if we require the agent to oscillate infinitely between the yellow, green, and red zones while avoiding the blue zone, we express this as $\varphi = \mathsf{GF}(y \wedge \mathsf{XF}(g \wedge \mathsf{XF}r)) \wedge \mathsf{G}\neg b$, which combines the properties of safety, reachability, and progress.

**From LTL to LDBA.** The satisfaction of LTL formulas can be formally defined using Limit Deterministic Büchi Automata (LDBAs). An LDBA can be derived from any LTL formula $\varphi$ and keeps track of the progression of $\varphi$ satisfaction [57].

**Definition 2.1** (Limit Deterministic Büchi Automaton (LDBA))**.** An LDBA is a tuple $\mathcal{L} = (\mathcal{B}, \Sigma \cup \mathcal{E}, P^{\mathcal{B}}, \mathcal{B}^{\star}, b_0)$, where $\mathcal{B}$ is a finite set of states, $\Sigma = 2^{AP}$ is an alphabet over atomic propositions, $P^{\mathcal{B}} : \mathcal{B} \times \Sigma \cup \mathcal{E} \rightarrow \mathcal{B}$ is a transition function, $\mathcal{B}^{\star} \subseteq \mathcal{B}$ is a set of accepting states, and $b_0 \in \mathcal{B}$ is the initial state. There ex-

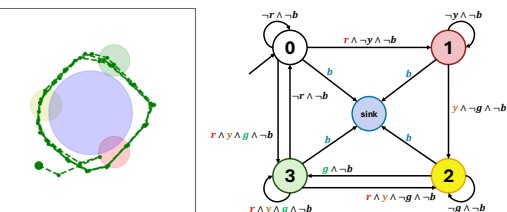

Figure 1: Left: FlatWorld Cycle environment with LTL spec $\varphi = \mathsf{GF}(y \wedge \mathsf{XF}(g \wedge \mathsf{XF}r)) \wedge \mathsf{G}\neg b$. Right: LDBA for $\varphi$ accepts paths reaching state 3 infinitely. Blue region $b$ leads to a sink.

ists a mutually exclusive partitioning of $\mathcal{B} = \mathcal{B}_D \cup \mathcal{B}_N$ such that $\mathcal{B}^{\star} \subseteq \mathcal{B}_D$ and for $(b, a) \in (\mathcal{B}_D \times \Sigma)$, then $P^{\mathcal{B}}(b, a) \subseteq \mathcal{B}_D$. $\mathcal{E}$ is a set of "jump" actions, also known as epsilon-transitions, for $b \in \mathcal{B}_N$ that transitions to $\mathcal{B}_D$ without evaluating any atomic propositions.

**Example.** In the *FlatWorld* environment (Fig. 1, left) and its corresponding LTL specification $\varphi = \mathsf{GF}(y \wedge \mathsf{XF}(g \wedge \mathsf{XF}r)) \wedge \mathsf{G}\neg b$, the LDBA is shown in Fig. 1 (right).

An infinite sequence of LDBA actions $(a_i)_{i=0}^\infty \in \Sigma^\infty$ induces a path $p = (b_i)_{i=0}^\infty$ according to $b_{i+1} = P^\mathcal{B}(b_i, a_i)$.

**Definition 2.2** (Limit Deterministic Büchi Automaton (LDBA)). An LDBA $\mathcal{L}$ accepts a path $(b_i)_0^\infty$ if and only if the path visits an accepting state $b^\star \in \mathcal{B}^\star$ infinitely often, that is: $\forall t \in \mathbb{N}, \exists t' > t$ such that $b_{t'} \in \mathcal{B}^\star$.

Given an LTL formula $\varphi$, we translate it into an LDBA $\mathcal{L}$. Satisfaction of $\varphi$ by an infinite sequence of $AP$ evaluations ($\omega$-word) corresponds directly to the acceptance of the path $(b_i)_{i=0}^\infty$ induced by the $\omega$-word in $\mathcal{L}$ (Def. 2.2).

## 2.2 Product MDP with LDBA

A Markov Decision Process (MDP) is defined as a tuple $\mathcal{M} = (S, A, P, \mu_0)$, where $S$ and $A$ represent the state and action spaces, which may be continuous or discrete. The transition kernel $P : S \times A \to \Delta(S)$ describes the system's dynamics, representing probability distributions over $S$. $\mu_0 \in \Delta(S)$ specifies the initial state distribution.

To integrate atomic propositions ($AP$s) into MDP states, as in previous work [63, 62, 6], we assume the existence of a labeling function $\mathcal{F} : S \to \Sigma$ which returns the atomic propositions that are true in that state. The labeling function can be thought of as a collection of event detectors that activate whenever the propositions in $AP$ are satisfied within the environment.

We define control policies over the product space of an MDP $\mathcal{M}$ and an LDBA $\mathcal{L}$, aiming to generate accepting trajectories that satisfy an LTL formula $\varphi$ from which the LDBA $\mathcal{L}$ is derived.

**Definition 2.3.** [Product MDP] Given MDP $\mathcal{M} = (S, A, P, \mu_0)$ and LDBA $\mathcal{L} = (\mathcal{B}, \Sigma \cup \mathcal{E}, P^\mathcal{B}, \mathcal{B}^\star, b_0)$, a product MDP $\mathcal{M}^\times = (S^\times, A^\times, P^\times, \mu_0^\times)$ synchronizes $\mathcal{M}$ with $\mathcal{L}$, where $S^\times = S \times \mathcal{B}$, $A^\times = A \cup \mathcal{E}$, and $\mu_0^\times(s, b) = \mu_0(s) \cdot 1_{b=b_0}$. The transition kernel $P^\times$ is defined using the labeling function $\mathcal{F}$ as follows:

$$P^\times((s', b') \mid (s, b), a) = \begin{cases} P(s' \mid s, a), & \text{if } a \in A, \ b' = P^\mathcal{B}(b, \mathcal{F}(s')) \\ 1, & \text{if } a \in \mathcal{E}, \ b' = P^\mathcal{B}(b, a), \ s = s' \\ 0, & \text{otherwise} \end{cases}$$

We say that $(s, b)$ is an LDBA-accepting product state if $b \in \mathcal{B}^*$. Def. 2.3 allows the connection of trajectories (and, consequently, policies) to the satisfaction of a given LTL formula. Consider an LTL formula $\varphi$ and its corresponding LDBA $\mathcal{L}$, along with a trajectory $\tau = (s_i, b_i)_{i=0}^\infty$ in the product MDP $\mathcal{M}^\times$. The trajectory $\tau \models \varphi$ (i.e., $\tau$ satisfies $\varphi$) if and only if $\mathcal{L}$ accepts the path $(b_i)_{i=0}^\infty$, the projection of $\tau$ onto the LDBA states.

Now, consider a policy $\pi : S^\times \to \Delta(A^\times)$. The probability that $\pi$ satisfies $\varphi$ can be expressed as the expected value of the indicator for trajectories satisfying $\varphi$: $P(\pi \models \varphi) = \mathbb{E}_{\tau \sim \pi}\left[1_{\tau \models \varphi}\right]$. Optimizing a policy $\pi$ for the satisfaction of an LTL specification $\varphi$ can therefore be formulated as finding $\pi^\star \in \arg\max_{\pi \in \Pi} P(\pi \models \varphi)$ where $\Pi$ denotes the space of all admissible policies.

## 2.3 Imitation Learning for LTL-Constrained Tasks

To maximize the probability of satisfying the LTL formula $P(\pi_\phi \models \varphi)$, an RL algorithm incentivize the agent to visit LDBA-accepting states as frequently as possible [27, 63]. However, this approach presents a significant challenge for exploration due to the inherent sparsity of the feedback: the agent receives rewards only upon making substantial progress toward task completion, such as reaching an accepting state in the LDBA. Visits to multiple non-accepting states within the LDBA during exploration may not provide meaningful learning signals, as there is no guidance from unexplored regions of the LDBA. **Our main idea** is to leverage the global structure of the LDBA and integrate it with expert task demonstrations to generate a dense reward signal along the LDBA paths for efficient agent exploration.

**Expert Demonstrations.** TiLoIL assumes expert demonstrations $\mathcal{D}_{\text{expert}}$ in the form of sequences of MDP states in $S$ observed during the execution of a task by an expert policy, providing information about the regions of the state space relevant for task completion. First, we do not assume that demonstrations are collected in the product MDP. This ensures that the discriminator cannot exploit LDBA state information as a shortcut to bias its decisions, which could result in uninformed learning

signals. Second, our demonstrations consist solely of state trajectories, reducing the burden of generating detailed action sequences. This choice also aligns with our goal of learning policies in the action space $A^\times$ of the product MDP. Since expert demonstrations operate within the raw MDP action space $A$, the demonstrated actions do not include the "jump" actions in $A^\times$.

**Imitation Learning.** The introduction of the Generative Adversarial Imitation Learning (GAIL) algorithm [29] has driven significant advances in scalable deep imitation learning methods [23, 24, 39, 34, 21, 8, 48]. Beyond adversarial approaches, several imitation learning algorithms aim to match the state action distributions of the expert and the agent through non-adversarial techniques, such as non-parametric models [38], random network distillation [64], support estimation [10], Wasserstein distance minimization [15], and moment matching [59].

In this paper, we adopt the GAIL framework. Recent studies [5] have shown that GAIL and its extensions consistently perform well at varying sample sizes. However, TiLoIL is not restricted to adversarial approaches and can be extended to other distribution matching imitation learning techniques. TiLoIL shapes reward signals using a discriminator $f_\psi(s)$, which is trained to distinguish between states from high-quality trajectories $B^+$—comprising expert demonstrations $\mathcal{D}_{\text{expert}}$—and states from low-quality trajectories $B^-$, consisting of the agent's own exploratory rollouts $\tau \sim \pi_\phi$ sampled from $\mathcal{M}^\times$. States that resemble those in $B^+$ receive higher rewards from the discriminator $f_\psi$, while those similar to $B^-$ receive lower rewards. TiLoIL employs an iterative training procedure, alternating between updating the discriminator $f_\psi$ and the policy $\pi_\phi$. The policy is optimized to generate trajectories that are increasingly indistinguishable from expert behavior, by maximizing the shaped reward $r_\psi((s,b),a) = \tanh(f_\psi(s))$. Given a discount factor $\gamma$, the policy objective is: $J_\pi(\phi) = \mathbb{E}_{\tau \sim \pi_\phi} \left[ \sum_{t=0}^\infty \gamma^t r_\psi((s_t, b_t), a_t) \right]$.

**Temporal Logic Imitation Learning**. Assuming access to expert demonstrations $\mathcal{D}_{\text{expert}}$, TiLoIL jointly optimizes the probabilistic satisfaction of an LTL formula $\varphi$ and the imitation learning objective $J_\pi$:

$$\pi^* = \arg \max_{\pi_\phi \in \Pi} \left( P(\pi_\phi \models \varphi), J_\pi(\phi) \right) \tag{1}$$

## 3  Imitation Learning with LTL Constraints

We emphasize that the learning objectives in our problem formulation, as presented in Eq. 1, are *not* in conflict. Among the set of policies that mimic expert demonstrations $\mathcal{D}_{\text{expert}}$ (i.e., maximize $J_\pi(\phi)$), there exists at least one policy $\pi_\phi$ that adheres to the specified LTL formula $\varphi$. Conversely, there exists one policy satisfying the specification $\varphi$ (i.e., $P(\pi_\phi \models \varphi)$) that also closely aligns with the behavior demonstrated by the expert. We formalize this intuition below:

**Theorem 3.1.** Let $\pi_1$ and $\pi_2$ be two policies with corresponding occupancy measures $\rho_{\pi_1}$ and $\rho_{\pi_2}$. For any Linear Temporal Logic (LTL) formula $\varphi$, the difference in the probabilities of satisfying $\varphi$ under these policies is bounded by twice the total variation distance between their occupancy measures:

$$|P[\pi_1 \models \varphi] - P[\pi_2 \models \varphi]| \leq 2D_{\text{TV}}(\rho_{\pi_1}, \rho_{\pi_2}), \tag{2}$$

where the total variation (TV) distance between the distributions $\rho_{\pi_1}$ and $\rho_{\pi_2}$ is given by $D_{\text{TV}}(\rho_{\pi_1}, \rho_{\pi_2}) = \frac{1}{2} \int_\tau |\rho_{\pi_1}(\tau) - \rho_{\pi_2}(\tau)| \, d\tau$.

Let $\pi_1$ be the expert policy $\pi_E$ from which demonstrations are collected, and $\pi_2$ be the imitation learning policy $\pi_\phi$. This theorem shows that minimizing the total variation distance between $\pi_E$ and $\pi_\phi$ leads to policies that maximize LTL satisfaction. The use of GAIL for imitation learning to minimize the JS (Jensen–Shannon) divergence reduces this TV distance, driving policies toward optimal satisfaction of LTL properties. The proof is in Appendix D.

**Main Challenge.** Theorem 3.1 does not directly apply in practice, as the infinite-length demonstrations required by LTL tasks are impractical to generate. For tasks with cyclic structures (e.g., Fig. 1), it is unrealistic to assume that expert demonstrations contain a large number of visits to LDBA-accepting states, due to the cost and complexity of constructing such demonstrations. Instead, we assume that demonstrations may include only one or two visits to an accepting state. This highlights a fundamental tension between the theoretical requirement of infinite visits to LDBA-accepting states and the inherently finite-horizon nature of expert demonstrations. First, the limited learning signal from such sparse demonstration data makes it challenging for the agent to generalize its behavior to satisfy LTL constraints over infinite horizons. Second, since environment states near

LDBA-accepting states are typically unique to expert demonstrations and differ substantially from the agent's exploratory behavior, particularly in the early stages of training, the discriminator may develop a reward function that assigns higher rewards to these states. Consequently, optimizing $J_\pi(\phi)$ under finite-length demonstrations could lead the agent to become trapped near an LDBA-accepting state, where the rewards are higher. In this way, the agent lacks incentive to actually reach the accepting state and then leaves it to initiate another trail aimed at reaching the accepting state again. This would result in suboptimal behavior, as the policy fails to reach LDBA-accepting states frequently enough.

## 3.1 Segmented Imitation

**Our core idea** to address the aforementioned main challenge is segmenting agent exploratory trajectories in $\mathcal{M}^\times$ based on visits to LDBA-accepting states, where each segment represents a subtrajectory toward an accepting state. From the agent's perspective, each segmented rollout can start in any state within the state space and the goal is to reach an LDBA-accepting state; when the agent reaches an accepting state, the next rollout starts directly from its current state, with the goal remaining the same: reaching an LDBA-accepting state. This structure incentivizes the agent to continuously seek LDBA-accepting states, regardless of the starting point of each trail, thus encouraging their infinite visits and improving $P(\pi_\phi \models \varphi)$ the probability of LTL formula satisfaction. Even if the demonstration contains only a single visit to an accepting state, it can be reused to guide the agent toward efficiently reaching LDBA-accepting states within each segment of its rollout, thereby optimizing $J_\pi(\phi)$. We formalize our method based on off-policy Q learning.

**Q-Function Update.** The Q-function $Q_\theta((s, b), a)$ for state and action pairs from $\mathcal{M}^\times$ is updated by minimizing the Bellman residual using data sampled from a sampled replay buffer $B$. The loss function is defined as follows:

$$J_Q(\theta) = \mathbb{E}_{((s,b),a,(s',b'))\sim B}\left[\frac{1}{2}\left(Q_\theta((s,b),a) - \hat{y}\right)^2\right], \tag{3}$$

The target value $\hat{y}$ is defined as:

$$\hat{y} = \begin{cases} 1/(1-\gamma), & b \in \mathcal{B}^\star \\ R((s,b)) + \gamma\mathbb{E}_{a'\sim\pi_\phi(\cdot|(s',b'))}\left[Q_{\texttt{targ}}\right], & b \notin \mathcal{B}^\star. \end{cases} \tag{4}$$

where $R((s, b)) = \tanh(f_\psi(s))$ is the reward from the learned discriminator function $f_\psi$ that separates $B^+$ as segmented expert demonstrations and $B^-$ as segmented policy rollouts. Here, $Q_{\texttt{targ}} = Q_{\bar{\theta}}((s', b'), a') - \alpha\log\pi_\phi(a'|(s', b'))$ is the target $Q$ value computed using a target network with parameters $\bar{\theta}$, $\alpha$ is the temperature parameter controlling the trade-off between reward and entropy, and $\log\pi_\phi(a'|(s', b'))$ is the entropy term used to encourage exploration.

Intuitively, we use the discriminator reward in $R((s, b))$ to train $Q_\theta$ to optimize $J_\pi(\phi)$ in our learning objectives in Eq. 1, encouraging the agent to mimic expert behavior to reach LDBA-accepting states. Upon reaching an LDBA-accepting state $(s, b)$ such that $b \in \mathcal{B}^\star$, we directly set the target value for $Q((s, b), a)$ (for any action $a$) to $\frac{1}{1-\gamma}$. First, this $Q$ value is sufficiently large to incentivize the agent to reach the LDBA-accepting state $(s, b)$, rather than lingering nearby, thereby ensuring continuous progress toward satisfying the LTL constraint. Second, in this way, each segmented rollout does not interfere with others, effectively addressing the main challenge.

**Policy ($\pi_\phi$) Update.** The policy $\pi_\phi(a|(s, b))$ is updated by minimizing the entropy-regularized expected $Q$-value, balancing exploration and exploitation. The loss is:

$$J_\pi(\phi) = \mathbb{E}_{(s,b)\sim B}\left[\mathbb{E}_{a\sim\pi_\phi(\cdot|(s,b))}\left[\alpha\log\pi_\phi(a|(s,b)) - Q_\theta((s,b),a)\right]\right]. \tag{5}$$

Here, the entropy term $\alpha\log\pi_\phi(a|(s, b))$ promotes exploration, while the Q-value term $-Q_\theta((s, b), a)$ encourages reward maximization.

The following theorem bounds the $Q$-function update in Eq. 3 relative to the optimal $Q$-value under the assumption of infinite-horizon expert demonstrations:

**Theorem 3.2.** Let $Q_\theta$ be the learned soft Q-function trained using a modified target value $Q^{\text{target}} = \frac{1}{1-\gamma}$ in accepting states $(s, b)$ where $b \in \mathcal{B}^\star$, and soft Bellman backups elsewhere. Assume $\pi_\phi(a \mid s, b) \propto \exp\left(\frac{1}{\alpha}Q_\theta((s, b), a)\right)$. Suppose that the reward function $R((s, b)) = \tanh(f_\psi(s))$ is bounded

by $R((s,b)) \in [R_{\min}, R_{\max}]$. Let $Q^*$ be the optimal $Q$-function under standard soft Bellman backups (without the modified target). Then, for any state-action pair $((s,b),a)$,

$$Q_\theta((s,b),a) - Q^*((s,b),a) \leq \gamma^{k(s,b)} \cdot \delta_{\max}$$

where $k(s,b)$ is the number of steps it takes from $(s,b)$ under $\pi_\phi$ to reach some accepting state $(s',b')$ with $b' \in \mathcal{B}^\star$, and $\delta_{\max} := \frac{1}{1-\gamma} - \frac{R_{\min} + \alpha \mathcal{H}_{\min}}{1-\gamma}$ is the worst-case overestimation error at any accepting states. $\mathcal{H}_{\min}$ denotes the minimum entropy of $\pi_\phi$.

The theorem shows that, while $Q$-values are inflated to make LDBA-accepting states attractive, this overestimation decays exponentially with distance. For states that appear early in a *segmented* trajectory—those farther from acceptance—the corresponding $Q$-values are not significantly overestimated. The learning process therefore remains grounded. The proof is in Appendix D.

## 3.2 Multi-Stage Discriminator Learning

The learning strategy described in Sec. 3.1 still faces challenges when long horizons are required to reach each LDBA-accepting state. We observe that many LTL tasks inherently consist of multiple stages. Take, for example, the FlatWorld Cycle task with the LTL specification $\varphi = \mathsf{GF}(y \wedge \mathsf{XF}(g \wedge \mathsf{XF}r)) \wedge \mathsf{G}\neg b$ and its LDBA, illustrated in Fig. 1. Any valid trajectory between the LDBA accepting states can be divided into three distinct stages. Initially, the LDBA state is 0 and the agent is in the white space for the first stage of reaching the red zone. Upon reaching the red zone $(r)$, the LDBA transitions from 0 to 1 and the agent is the second stage of reaching the yellow zone. If the agent touches the blue region in the middle at any point, the LDBA transitions into a sink state and remains there for the rest of the episode. *How can we design a staged approach for LDBAs that makes the learning process more efficient compared to treating them as a single monolithic stage?*

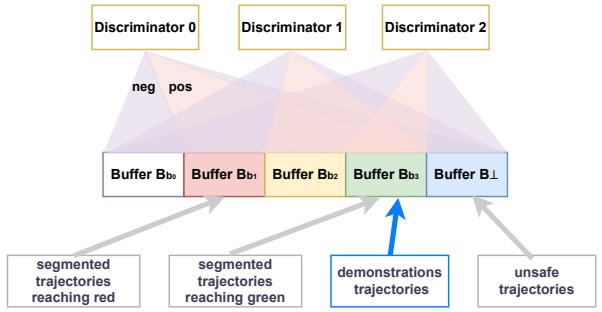

Figure 2: We illustrate the multistage discriminator for the FlatWorld Cycle environment in Fig.1. TiLoIL maintains separate stage buffers to store trajectories corresponding to different LDBA states. Here, the colors of the buffers match the corresponding automaton states in Fig. 1. Each segmented trajectory is assigned to only one stage buffer based on its maximal stage (Eq. 7). $B_{b_0}$ holds segmented trajectories that do not enter any colored zones, while $B_{b_1}$ stores segmented trajectories that visit the red region but do not reach yellow. For Discriminator 0, negative data are drawn from buffer $B_{b_0}$ and buffer $B_\perp$ which stores unsafe trajectories, while positive data are sampled from buffers $B_{b_1}$, $B_{b_2}$, and $B_{b_3}$. As such, the discriminator provides reward signals that encourage the agent to reach the red region and beyond.

We begin by presenting key definitions, followed by an illustration of our proposed approach. Define $b_i \rightsquigarrow b_j$ as an acyclic path in the (graph representation of) LDBA $\mathcal{L}$, where $b_i$ and $b_j$ are states in $\mathcal{L}$, such that the path begins at $b_i$, ends at $b_j$, and *does not include any accepting states* $b_f \in \mathcal{B}^*$ other than possibly $b_j$ if $b_j$ is an accepting state:

$b_i \rightsquigarrow b_j \iff \exists$ path $(b_i, b_{i+1}, \ldots, b_j)$ in $\mathcal{L}$ such that

$$b_k \notin \mathcal{B}^* \; \forall k \neq j, \text{ and } \forall k, \ell, \; k \neq \ell \implies b_k \neq b_\ell. \quad (6)$$

Define $b_i \rightsquigarrow^* b_j \iff (b_i \rightsquigarrow b_j \vee b_i = b_j)$. We define a sink state of an LDBA $\mathcal{L}$ as a state $b_s \in \mathcal{B}$ such that $P^{\mathcal{B}}(b_s, \cdot) = b_s$. Once the agent transitions to a sink state, it cannot escape, thereby failing to reach any accepting states. Sink states are useful for modeling safety properties, such as globally avoiding the blue obstacle shown in Fig. 1. We use $\mathrm{SINK}(\mathcal{L})$ to denote all sink states of $\mathcal{L}$.

**Multistage Discriminators.** In a dense reward setting for multistage tasks, the reward of an environment state associated with an LDBA state $b_l$ should exceed that of $b_k$ if $b_k \rightsquigarrow b_l$, as this encourages the agent to progress toward the LDBA's accepting states. If each state in an LDBA $\mathcal{L}$ is viewed as an individual stage, a separate discriminator can be trained for each stage to serve as a dense reward for that particular stage. By training stage-specific discriminators, we can effectively

guide the agent's progress through the different stages of the task. To train the discriminators for different stages, we establish positive and negative data for each discriminator. We assign a maximal stage to each trajectory $\tau$, which is determined as the LDBA state that advanced the furthest towards accepting states among all LDBA states within $\tau$:

$$\text{MaxStage}(\tau : ((s_0, b_0), \dots, (s_N, b_N))) = b_i \text{ such that } \exists \text{ path } \rho = (b_i \rightsquigarrow b_f) \wedge b_f \in \mathcal{B}^*.$$
$$\forall b_j \in \{b_0, \dots, b_N\} \setminus b_i, \ b_j \text{ not in } \rho \quad (7)$$

For the discriminator associated with the LDBA state $b_k$, positive data include trajectories $\tau^+$ with maximal stage $\text{MAXSTAGE}(\tau^+)$ progressing *beyond* $b_k$ and *up to* an accepting state, formally expressed as $\exists b_f \in \mathcal{B}^*. \ b_k \rightsquigarrow \text{MAXSTAGE}(\tau^+) \rightsquigarrow^* b_f$. Conversely, negative data consist of trajectories $\tau^-$ that only reach up to $b_k$, such that $\text{MAXSTAGE}(\tau^-) \rightsquigarrow^* b_k$, or hit any sink state $\text{MAXSTAGE}(\tau^-) \in \text{SINK}(\mathcal{L})$. *An example is given in Fig. 2.*

Once the positive and negative data for each discriminator for an LDBA state $b_k$ have been established, we train a discriminator $f_{b_k}$ that predicts if an MDP state $s \in S$ comes from the positive trajectories whose maximal stage is beyond the LDBA state $b_k$, as opposed to the negative trajectories that fail to progress beyond $b_k$ or trap into sink states. The discriminator is trained using the BCE loss where positive data $B^+$ includes states from $\tau^+$ trajectories and negative data $B^-$ consists of states from $\tau^-$ trajectories. We note that $B^+$ always includes (segmented) expert demonstrations that surpass all LDBA states on the path toward accepting states.

**Multistage Reward Formulation.** The next step is to combine these discriminators to create a reward function that guides the agent to LDBA-accepting states. Our formulation is inspired by [45]. We define our learned reward function for a product MDP state in a multi-stage task as follows:

$$R((s, b)) = \frac{\text{SIDX}(b) + \beta \cdot \tanh(f_b(s))}{\mathcal{N}(b)} \quad (8)$$

where $\text{SIDX}(b)$ computes the length of the longest acyclic path in the LDBA from the initial state to $b$, serving as an approximation of the stage index of the product MDP state $(s, b)$, and $\beta$ is a hyperparameter. The $\tanh$ function is used to bound the output of the discriminators. As the range of the $\tanh$ function is $(-1, 1)$, any $\beta < \frac{1}{2}$ ensures that the reward of a state in stage $k+1$ is always higher than that of stage $k$. In practice, we use $\beta = \frac{1}{3}$. Dividing it by $\mathcal{N}(b)$, where $\mathcal{N}(b)$ denotes the length of the longest acyclic path in the LDBA from the initial state to any accepting state through $b$, scales the rewards to values less than 1.

**Main Algorithm.** The main algorithm of TiLoIL is summarized in Algorithm 1. We use SAC [25] for policy training. In addition to the regular replay buffer $B$ used in SAC, TiLoIL maintains different stage buffers $B_{b_k}$ to store trajectories corresponding to different LDBA states $b_k$. Each trajectory is assigned to only one stage buffer based on its maximal stage (Eq. 7). During the training of the discriminators, we sample data from the union of multiple buffers. Policy updates are then guided by reward functions derived from these trained discriminators.

## 4  Experiments

This section empirically evaluates TiLoIL by addressing the following questions: (Q1) Does TiLoIL improve exploration in LTL-constrained tasks? (Q2) Are the new learning-from-demonstrations strategies in TiLoIL necessary to learn policies that align with the LTL constraints?

**Baselines.** To answer Q1, we compare TiLoIL with three state-of-the-art policy optimization algorithms for LTL task objectives: (1) LCER [63], a counterfactual experience replay scheme, (2) Cycler [55], a method focusing on cycle environments, and (3) DRL$^2$ [6], a direct exploration algorithm that encodes LDBAs as a Markov reward process for reward shaping. This evaluation focuses on TiLoIL's exploration capabilities relative to these RL-based approaches. For Q2, we compare TiLoIL with conventional imitation learning algorithms GAIL [29], PWIL [14] and SQIL [53]. As discussed in Sec. 2.3, TiLoIL can be integrated into any generative adversarial and distribution matching imitation learning methods. However, rather than evaluating a broad range of algorithms, our focus is on assessing how our proposed strategies improve the imitation learning algorithm that it builds on (GAIL). In our experiments, we combined SAC [25] with TiLoIL, GAIL, PWIL, and SQIL to perform policy updates on batches of data sampled from the agent's replay buffer.

**Algorithm 1** TiLoIL Main Algorithm

---

**Require:** MDP $\mathcal{M}$, LDBA $\mathcal{L} = (\mathcal{B}, \Sigma \cup \mathcal{E}, P^{\mathcal{B}}, \mathcal{B}^{\star}, b_0)$, Demonstration dataset $\mathcal{D} := \{\tau^0, \tau^1, \ldots\}$
1: Initialize policy $\pi_\phi$, critic $Q_\theta$, replay buffer $B$
2: Initialize discriminators $f_{b_0}, f_{b_1}, \ldots$ for $b_k \in \mathcal{B} \setminus \mathcal{B}^*$
3: Initialize stage buffers $B_{b_0}, B_{b_1}, \ldots$ for $b_k \in \mathcal{B}$
4: Populate the stage buffers for $\mathcal{B}^*$ with $\mathcal{D}$
5: **for** each iteration **do**
6:   Sample trajectories $\mathcal{T}$ by executing $\pi_\phi$ in $\mathcal{M}^{\times}$ $\mathbin{/\!/} \mathcal{M}^{\times} \equiv M \times \mathcal{L}$
7:   Segment $\mathcal{T}$ at LDBA-accepting states $(s, b)$ such that $b \in \mathcal{B}^*$ to obtain $\{\tau_\pi^{0\times}, \tau_\pi^{1\times}, \ldots\}$
8:   **for** each trajectory $\tau_\pi^{i\times}$ in $\{\tau_\pi^{0\times}, \tau_\pi^{1\times}, \ldots\}$ **do**
9:     $b_j \leftarrow \text{MAXSTAGE}(\tau_\pi^{i\times})$
10:    **if** $b_j \in \text{SINK}(\mathcal{L})$ **then**
11:      $B_\perp \leftarrow B_\perp \cup \{\tau_\pi^i\}$
12:    **else**
13:      $B_{b_j} \leftarrow B_{b_j} \cup \{\tau_\pi^i\}$
14:   $B \leftarrow B \cup \{\tau_\pi^{0\times}, \tau_\pi^{1\times}, \ldots\}$
15:   **for** each gradient step for the discriminators **do**
16:     **for** each state $b_k$ in $\mathcal{B} \setminus \mathcal{B}^* \setminus \text{SINK}(\mathcal{L})$ **do**
17:       Sample positive data from $\bigcup B_{b_i}$ for all $b_i$ s.t. $\exists b_f \in \mathcal{B}^*.\ b_k \rightsquigarrow b_i \rightsquigarrow^* b_f$
18:       Sample negative data from $\bigcup B_{b_i} \cup B_{b_k} \cup B_\perp$ for all $b_i$ s.t. $b_i \rightsquigarrow^* b_k$
19:       Update $f_{b_k}$ using BCE loss
20:   **for** each gradient step for the policy $\pi_\phi$ **do**
21:     Update $Q_\theta$ via Eq. 3 and $\pi_\phi$ via Eq. 5 by SAC via samples from $B$

---

**Environments and Tasks.** Our benchmarks, as visualized in Fig. 3, include tasks drawn from LCER [63] and DRL[2] [6], complemented by new environments we developed to highlight exploration challenges in LTL tasks. On the top left, *GridCircular Hard* is a discrete 2D grid world where the agent moves in four cardinal directions or stays still[1]. The environment is a cross-shaped grid of five squares, where the center is an obstacle. The agent must repeatedly loop through the outer squares while avoiding the center. Next in the upper row, a point agent in *FlatWorld Stabilization* must stabilize in the yellow zone, while in *FlatWorld Cycle*, it oscillates between red, yellow, and green infinitely, always avoiding blue regions. On the left of the second row, the Doggo agent, a 12-DoF quadruped robot [52], must (i) traverse a narrow corridor collision-free (*Doggo Avoid*) and (ii) sequentially navigate two designated zones (*Doggo Navigate*). In the bottom-left, a Fetch robotic arm [17] performs four tasks: (i) guiding its gripper to a target position while minimizing lateral movements (*Fetch Avoid*); (ii) achieving horizontal alignment of three cubes (*Fetch Align*); (iii) placing a block into a tray when the tray is present, or into the goal region when it is absent (*FetchPlace Tray*); and (iv) repeatedly moving the cube to the current goal region (red) and pressing the green button to change the goal location, requiring the agent to alternate between goal reaching and button pressing (*FetchPlace Button*). On the right, in the *Carlo* environment, the agent drives a self-driving simulator based on a bicycle model counterclockwise on a circular track, repeatedly visiting two blue regions without crashing. In *Cheetah Flip*, the HalfCheetah agent performs frontflips to alternate between standing on its front and back legs infinitely. In *SixteenRooms*, the agent is initially positioned at the center of the bottom-left room. The task requires the agent to follow a circular path indefinitely using one of two possible routes in a grid of rooms: a large loop passing through a set of rooms and a smaller loop passing through another subset. The agent must avoid collisions with walls separating rooms and learn to repeatedly traverse one of the valid loops. All task specifications in LTL and additional environment details are provided in Appendix H.

**Demonstrations.** While TiLoIL assumes (finite-length) expert demonstrations, in evaluation, we relax this assumption and use demonstrations that are not necessarily optimal. For our tasks, the demonstrations are generated by designing dense rewards and training individual policies for each stage of a task (e.g., training a policy to reach the yellow zone and another to reach the green zone in *FlatWorld Cycle*). Trajectories that successfully reach accepting states are collected by sequentially executing these policies. Both TiLoIL and the baselines—GAIL, PWIL, and SQIL—are provided

---

[1]We compared TiLoIL with the baselines in a suit of discrete environments from [6] in Appendix E.

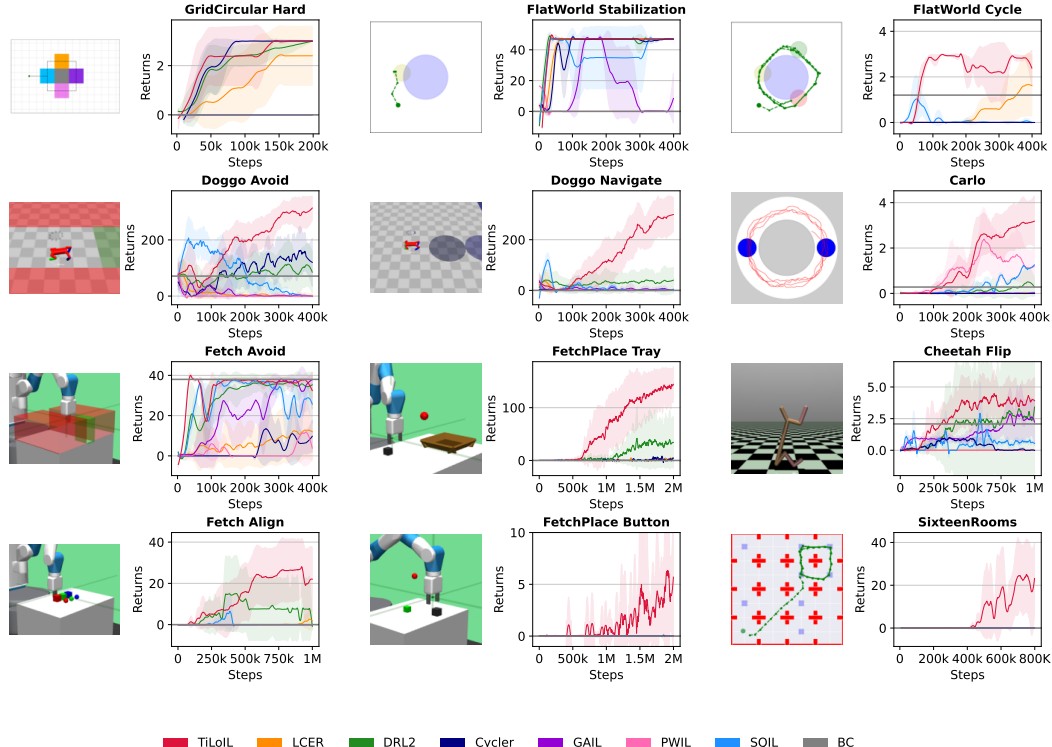

Figure 3: Returns under eventual discounting [63] comparing TiLoIL and the baselines over 10 random seeds. The shaded region indicates the standard deviation.

with only **5 demonstrations**. In environments with cyclic structures—*GridCircular*, *FlatWorld Cycle*, *Carlo*, *Cheetah Flip*, *FetchPlace Button*, and *SixteenRooms*—each trajectory contains at most **2 visits to LDBA accepting states**. Our goal is to show that even finite-length, non-optimal demonstrations—obtained from suboptimal policies that visit LDBA-accepting states only once or twice—can be effectively used to learn policies satisfying LTL objectives over infinite horizons.

**Results.** In Fig. 3, the x-axis shows the environment steps and the y-axis shows the cumulative returns under eventual discounting (App. H.3). Computing the probability of LTL satisfactions over infinite horizons requires estimating policy occupancies, which is intractable. This return function, as a proxy for the likelihood of task satisfaction [63], counts visits to LDBA-accepting states, assigning a reward of 1 per visit with future discounts applied only at such states.

(Q1) Does TiLoIL help the learning process for LTL-constrained tasks? According to Fig. 3, TiLoIL significantly accelerates learning compared to the RL-based policy optimization approaches LCER, DRL$^2$ and Cycler. In our experience, Cycler faces challenges in scaling to high-dimensional continuous environments. Integrating the global structure of an LDBA and with expert task demonstrations, TiLoIL provides informative rewards in each LDBA transition, effectively guiding exploration toward accepting states, an advantage absent in LCER. Although DRL$^2$ also learns an intrinsic reward signal along LDBA paths to aid exploration, TiLoIL provides a stronger guidance through demonstrations, allowing faster transitions to LDBA-accepting states, particularly in complex environments, e.g. *Fetch Align* and *Carlo*. In *Carlo*, TiLoIL completes 3–4 rounds within the track in 500 steps, while the baselines struggle to complete even one.

(Q2) Are segmented imitation and multistage discriminator learning in TiLoIL necessary? Our results show that TiLoIL generates a significant lift over its baseline algorithms GAIL, PWIL and SQIL in the learning curves in Fig. 3 for all tasks with cyclic structures, such as *Cheetah Flip* and *FetchPlace Button*, highlighting the importance of segmented imitation for generalizing policies to repeated accepting-state visits over infinite horizons. In other multistage tasks, such as *Doggo Navigate*, *Fetch Align*, and *SixteenRooms*, the improvements over these imitation learning baselines are attributed to the generation of stage-specific rewards that guide intermediate goal achievement, without which the learning signal from a monolithic discriminator may lead to a flat value landscape.

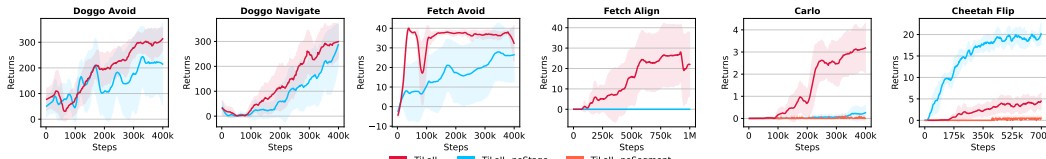

Figure 4: Ablation studies for TiLoIL over 10 random seeds using returns under eventual discounting. Results for TiLoIL-noSegment are omitted in *Doggo* and *Fetch* tasks. For these non-circular tasks, their LDBAs absorb at accepting states when safety is ensured. Thus, trajectory segmentation has negligible impact, leading to similar reward curves for TiLoIL-noSegment and TiLoIL in these tasks.

**Ablation Studies.** We assess the individual contributions of (a) segmented imitation and (b) multistage discriminator learning to the overall performance. The first ablation, TiLoIL-noSegment, only performs part (b) for multistage discriminator learning. The second ablation TiLoIL-noStage only performs part (a) for segmented imitation. Fig. 4 confirms the importance of segmented imitation, as circular tasks (*Carlo* and *Cheetah Flip*) cannot be solved without it. Multistage discriminator learning is also crucial to ensure sample-efficient learning in all tasks except *Cheetah Flip*. TiLoIL-noStage performs extremely well in this challenging task. We found that the multistage discriminator in TiLoIL incorporates successful trajectories that barely achieve standing on the back legs, sometimes leading the agent to just satisfy this subtask without generating sufficient momentum to round over.

In our reward formulation (Eq. 8), $\beta$ controls the contribution of shaped rewards from multi-stage discriminators. As shown in the ablation results in Appendix F.1, a larger $\beta$ yields improved performance on complex tasks, as the shaped rewards from the discriminator provide effective guidance for learning. Additional ablations in Appendix F.2 and Appendix F.3 further examine TiLoIL's scalability to complex LTL formulas and its robustness to incomplete or unsafe demonstrations.

We experiment with TiLoIL under different numbers of demonstrations and observed that it can effectively bootstrap learning without requiring many demonstrations. See Appendix G for details.

## 5 Related Work

Reinforcement learning from linear temporal logic (LTL) has advanced significantly, with many approaches leveraging structural insights to guide learning [54, 18, 30, 31, 12, 13]. Early methods aligned value and policy optimization using product MDPs and reward signals to encourage task satisfaction [9, 12, 13, 27, 28, 37, 43], leading to principled approaches that optimize lower bounds on formula satisfaction [56, 63]. To address reward sparsity, several methods emerged: rule-based approaches [43], accepting frontier function [27, 28], rewards for initial visits [11] or adapted annotated maps [67]. Automata structure has also been used to learn hierarchical [28, 31, 35], goal-conditioned [51], or modular [11] policies. Recent works further improve LTL-guided exploration via meta-learning [41, 62], temporal reward shaping [6], eventual discounting [63] and cyclic temporal constraints for recurring goals [55]. We discuss related work in broader contexts in Appendix B.

## 6 Conclusion

We present TiLoIL, an imitation learning framework for policy optimization under LTL constraints. TiLoIL employs segmented imitation learning to guide the agent toward LDBA-accepting states following task demonstrations from any state within the state space, thus optimizing the satisfaction of the LTL formula over infinite horizons. Furthermore, TiLoIL integrates LDBA structures with task demonstrations to construct dense reward signals along LDBA paths, facilitating efficient exploration toward accepting states. Our results demonstrate that TiLoIL effectively solves various high-dimensional tasks with limited demonstrations, outperforming various baselines.

**Limitations.** While TiLoIL demonstrates strong performance in leveraging demonstrations to satisfy LTL objectives, effectively exploiting suboptimal or incomplete demonstrations remains challenging. We hypothesize that integrating more exploratory RL algorithms could mitigate this limitation. Future work should focus on combining TiLoIL's exploitation mechanism with exploration strategies that exploit LDBA structure for more efficient learning of LTL-constrained policies.

## Reproducibility Statement

The code for TiLoIL is available on `https://github.com/RU-Automated-Reasoning-Group/TiLoIL`.

## Acknowledgements

We thank the anonymous reviewers for their comments and suggestions. This work was supported by NSF Award #CCF-2124155.

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

## A Impact Statement

This paper presents work whose goal is to advance the field of Machine Learning. There are many potential societal consequences of our work, none of which we feel must be specifically highlighted here.

## B Related Work Discussion

**Learning from demonstrations.** Learning from demonstration is particularly valuable when designing a reward function is challenging. It allows agents to learn desired behaviors by observing and mimicking expert demonstrations. Some methods utilize classification-based rewards, where a reward function is trained by classifying goals [58, 36, 19] or by categorizing demonstration trajectories [69]. However, these rewards are trained exclusively on offline datasets, making them vulnerable to exploitation by a reinforcement learning agent.

Unlike multi-task RL approaches for LTL [44, 60, 4, 33] that define subtasks for zero-shot generalization to long-horizon or unseen tasks, TiLoIL focuses on mitigating the inherent exploration challenges in LTL-guided tasks, making it orthogonal to these approaches. Compared to approaches that require large-scale offline pretraining for LTL tasks [20], TiLoIL learns effectively from a small number of expert trajectories. Temporal Logic Imitation [65] integrates high-level LTL planning with pre-existing low-level controllers for continuous motion execution, whereas TiLoIL jointly learns both in an end-to-end manner. [32] transforms LTL specifications into a differentiable loss function, while TiLoIL addresses tasks where the reward signal from LTL remains sparse where such a differentiable loss approach is infeasible.

**Inverse reinforcement learning.** The above issue can be solved by Inverse Reinforcement Learning. Inverse Reinforcement Learning (IRL) is a crucial tool in learning from demonstrations [1, 47]. It aims at uncovering the underlying reward function from observed behaviors, which is particularly useful in scenarios where reward structures are not explicitly defined. Recently, Adversarial Imitation Learning (AIL) [29, 39, 24, 23]methods have been introduced, functioning in a manner akin to Generative Adversarial Networks (GANs). In these approaches, a generator (the policy) is trained to maximize the confusion of a discriminator, while the discriminator, acting as a surrogate for the reward function, is trained to differentiate between the agent's trajectories and the expert demonstrations. The introduction of GAIL [29] has driven significant advances in scalable deep imitation learning methods [23, 24, 39, 34, 21, 8, 48]. Beyond adversarial approaches, several imitation learning algorithms aim to match the state action distributions of the expert and the agent through non-adversarial techniques, such as non-parametric models [38], random network distillation [64], support estimation [10], Wasserstein distance minimization [15], and moment matching [59].

**Rank reward learning** The above issue could be solved by decomposing the tasks into easier sub-tasks. Hierarchical Reinforcement Learning (HRL) [22, 46, 42] methods decompose policies into sub-policies, each designed to address specific sub-tasks. Some approaches learn rewards with underlying substructures. DrS [45]decompose tasks by stage indicators, rank2reward [68] by learning videos; some methods [6, 63, 55]combined Buchi Automaton with the problem of labeling subtasks.

## C Linear Temproal Logic

**Syntax of LTL** The syntax of Linear Temporal Logic (LTL) is defined over a set of atomic propositions AP. A state labeling function $\mathcal{F} : S \rightarrow \Sigma$ maps each state $s \in S$ to a subset of atomic propositions $\Sigma = 2^{\mathsf{AP}}$, where $\Sigma$ is the alphabet formed by the powerset of AP. An LTL formula $\varphi$ is constructed using the following grammar:

$$\varphi ::= \mathsf{true} \mid \mathsf{false} \mid p \mid \neg\varphi \mid (\varphi \wedge \psi) \mid (\varphi \vee \psi) \mid \mathsf{X}\varphi \mid \mathsf{F}\varphi \mid \mathsf{G}\varphi \mid (\varphi\mathsf{U}\psi)$$

where $p \in \mathsf{AP}$, and $\varphi$ and $\psi$ are LTL formulas. The logical connectives true, false, $\neg$ (negation), $\wedge$ (conjunction), and $\vee$ (disjunction) are standard. The temporal operators are: $\mathsf{X}$ for "next", $\mathsf{F}$ for "eventually", $\mathsf{G}$ for "globally", and $\mathsf{U}$ for "until".

**Semantics of LTL** The semantics of LTL is defined over infinite sequences of states $\tau = s_0, s_1, s_2, \ldots$, where each state $s_i \in S$ is associated with a set of atomic propositions $\mathcal{F}(s_i) \subseteq \mathsf{AP}$.

We write $\tau, i \models \varphi$ to indicate that $\varphi$ holds at position $i$ of the sequence $\tau$. The semantics are given inductively as follows:

$$\tau, i \models \text{true} \quad \text{(always holds)}$$
$$\tau, i \not\models \text{false} \quad \text{(never holds)}$$
$$\tau, i \models p \quad \Longleftrightarrow \quad p \in \mathcal{F}(s_i) \quad \text{for } p \in \text{AP}$$
$$\tau, i \models \neg\varphi \quad \Longleftrightarrow \quad \tau, i \not\models \varphi$$
$$\tau, i \models (\varphi \wedge \psi) \quad \Longleftrightarrow \quad \tau, i \models \varphi \text{ and } \tau, i \models \psi$$
$$\tau, i \models (\varphi \vee \psi) \quad \Longleftrightarrow \quad \tau, i \models \varphi \text{ or } \tau, i \models \psi$$
$$\tau, i \models \mathsf{X}\varphi \quad \Longleftrightarrow \quad \tau, i+1 \models \varphi$$
$$\tau, i \models \mathsf{F}\varphi \quad \Longleftrightarrow \quad \exists j \geq i, \tau, j \models \varphi$$
$$\tau, i \models \mathsf{G}\varphi \quad \Longleftrightarrow \quad \forall j \geq i, \tau, j \models \varphi$$
$$\tau, i \models (\varphi\mathsf{U}\psi) \quad \Longleftrightarrow \quad \exists j \geq i, (\tau, j \models \psi \text{ and } \forall k \in [i, j), \tau, k \models \varphi).$$

In English, the semantics of Linear Temporal Logic (LTL) are defined over infinite sequences of states $\tau = s_0, s_1, s_2, \ldots$, where each state $s_i$ represents a snapshot of the system. A state labeling function $\mathcal{F} : S \to \Sigma$ assigns a set of atomic propositions (AP) to each state, indicating which propositions are true. The satisfaction of an LTL formula is evaluated along these sequences. For example, $p$ holds at a state $s_i$ if $p \in \mathcal{F}(s_i)$, while $\mathsf{X}p$ requires $p$ to hold in the next state $s_{i+1}$. Temporal operators like $\mathsf{F}p$ ("eventually $p$") and $\mathsf{G}p$ ("globally $p$") extend this reasoning over future states. Similarly, $\varphi\mathsf{U}\psi$ ("$\varphi$ until $\psi$") requires $\varphi$ to hold continuously until $\psi$ becomes true at some future state.

For instance, the formula $\mathsf{F}y$ expresses that the system must eventually reach a state where $y$ holds, which could represent a robot reaching a target area. The formula $\mathsf{G}\neg b$ specifies that $b$, such as a hazardous condition, must always be avoided. More complex behaviors can also be modeled, such as $\mathsf{G}(\mathsf{F}r)$, which ensures the system repeatedly visits states where $r$ is true, or $\mathsf{G}(p\mathsf{U}q)$, where $p$ must hold until $q$ becomes true. These examples demonstrate how LTL can express diverse temporal properties for dynamic systems.

## D  Proofs of Theorem 3.1 and 3.2

**Theorem 3.1** Let $\pi_1$ and $\pi_2$ be two policies with corresponding occupancy measures $\rho_{\pi_1}$ and $\rho_{\pi_2}$. For any Linear Temporal Logic (LTL) formula $\varphi$, the difference in the probabilities of satisfying $\varphi$ under these policies is bounded by twice the total variation distance between their occupancy measures:

$$|P[\pi_1 \models \varphi] - P[\pi_2 \models \varphi]| \leq 2D_{\text{TV}}(\rho_{\pi_1}, \rho_{\pi_2}), \tag{9}$$

where the total variation (TV) distance between the distributions $\rho_{\pi_1}$ and $\rho_{\pi_2}$ is given by

$$D_{\text{TV}}(\rho_{\pi_1}, \rho_{\pi_2}) = \frac{1}{2} \int_\tau |\rho_{\pi_1}(\tau) - \rho_{\pi_2}(\tau)| \, d\tau. \tag{10}$$

*Proof.* The probability of satisfying $\varphi$ under policy $\pi_1$ can be expressed as an expectation:

$$P[\pi_1 \models \varphi] = \mathbb{E}_{\tau \sim \rho_{\pi_1}}[\mathbf{1}(\tau \models \varphi)] = \int_\tau \mathbf{1}(\tau \models \varphi)\rho_{\pi_1}(\tau)d\tau. \tag{11}$$

Similarly, for policy $\pi_2$:

$$P[\pi_2 \models \varphi] = \mathbb{E}_{\tau \sim \rho_{\pi_2}}[\mathbf{1}(\tau \models \varphi)] = \int_\tau \mathbf{1}(\tau \models \varphi)\rho_{\pi_2}(\tau)d\tau. \tag{12}$$

Thus, the absolute difference between these probabilities is:

$$|P[\pi_1 \models \varphi] - P[\pi_2 \models \varphi]| = \left| \int_\tau \mathbf{1}(\tau \models \varphi)(\rho_{\pi_1}(\tau) - \rho_{\pi_2}(\tau))d\tau \right|. \tag{13}$$

Applying the triangle inequality:

$$|P[\pi_1 \models \varphi] - P[\pi_2 \models \varphi]| \leq \int_\tau |\rho_{\pi_1}(\tau) - \rho_{\pi_2}(\tau)| \, d\tau. \tag{14}$$

By the definition of total variation distance,

$$\int_\tau |\rho_{\pi_1}(\tau) - \rho_{\pi_2}(\tau)| \, d\tau = 2D_{\text{TV}}(\rho_{\pi_1}, \rho_{\pi_2}). \tag{15}$$

Thus, we obtain the desired bound:

$$|P[\pi_1 \models \varphi] - P[\pi_2 \models \varphi]| \leq 2D_{\text{TV}}(\rho_{\pi_1}, \rho_{\pi_2}). \tag{16}$$

This completes the proof. $\square$

**Theorem 3.2** Let $Q_\theta$ be the learned soft $Q$-function trained using a modified target value $Q^{\text{target}} = \frac{1}{1-\gamma}$ in accepting states $(s, b)$ where $b \in \mathcal{B}^\star$, and soft Bellman backups elsewhere. Assume $\pi_\phi(a \mid s, b) \propto \exp\left(\frac{1}{\alpha} Q_\theta((s, b), a)\right)$. Suppose that the reward function $R((s, b)) = \tanh(f_\psi(s))$ is bounded by $R((s, b)) \in [R_{\min}, R_{\max}]$. Let $Q^*$ be the optimal $Q$-function under standard soft Bellman backups (without the modified target). Then, for any state-action pair $((s, b), a)$,

$$Q_\theta((s, b), a) - Q^*((s, b), a) \leq \gamma^{k(s,b)} \cdot \delta_{\max}$$

where $k(s, b)$ is the number of steps it takes from $(s, b)$ under $\pi_\phi$ to reach some accepting state $(s', b')$ with $b' \in \mathcal{B}^\star$, and $\delta_{\max} := \frac{1}{1-\gamma} - \frac{R_{\min} + \alpha \mathcal{H}_{\min}}{1-\gamma}$ is the worst-case overestimation error at any accepting states. $\mathcal{H}_{\min}$ denotes the minimum entropy of $\pi_\phi$.

*Proof.* We prove the result by backward induction along the trajectory, using the soft Bellman equation and the definition of $\delta_{\max}$.

**Base case:** At the final step $k = k(s, b)$, the overestimation is at most:

$$\delta_{\max} = \frac{1}{1-\gamma} - \left(\frac{R_{\min} + \alpha \mathcal{H}_{\min}}{1-\gamma}\right),$$

**Inductive step:** Suppose at step $i + 1$ we have:

$$Q_\theta((s_{i+1}, b_{i+1}), a_{i+1}) \leq Q^*((s_{i+1}, b_{i+1}), a_{i+1}) + \gamma^{k-(i+1)} \delta_{\max}.$$

Then for step $i$:

$$
\begin{aligned}
Q_\theta((s_i, b_i), \, a_i) &= \\
&= R((s_i, b_i)) + \gamma \, \mathbb{E}_{s_{i+1}, b_{i+1}} \left[ \mathbb{E}_{a' \sim \pi_\phi(\cdot | (s_{i+1}, b_{i+1}))} \left[ Q_\theta((s_{i+1}, b_{i+1}), a') \right] \right] \\
&\leq R((s_i, b_i)) + \gamma \, \mathbb{E}_{s_{i+1}, b_{i+1}} \left[ \mathbb{E}_{a' \sim \pi_\phi(\cdot | (s_{i+1}, b_{i+1}))} [Q^*((s_{i+1}, b_{i+1}), a') + \gamma^{k-(i+1)} \delta_{\max}] \right] \\
&\leq R((s_i, b_i)) + \gamma \, \mathbb{E}_{s_{i+1}, b_{i+1}} \left[ \mathbb{E}_{a' \sim \pi_*(\cdot | (s_{i+1}, b_{i+1}))} [Q^*((s_{i+1}, b_{i+1}), a') + \gamma^{k-(i+1)} \delta_{\max}] \right] \\
&= Q^*((s_i, b_i), a_i) + \gamma^{k-i} \delta_{\max}.
\end{aligned}
$$

where $\pi^*$ is the optimal policy under $Q^*$.

Thus, the overesitmation error at $(s, b)$ can propagate at most $\gamma^{k(s,b)} \cdot \delta_{\max}$:

$$Q_\theta((s, b), a) \leq Q^*((s, b), a) + \gamma^{k(s,b)} \cdot \delta_{\max}.$$

$\square$

Even though our method clips the Q-value at accepting states to a large constant (e.g., $\frac{1}{1-\gamma}$), the theoretical bound shows:

$$Q_\theta((s, b), a) - Q^*((s, b), a) \leq \gamma^{k(s,b)} \cdot \delta_{\max}$$

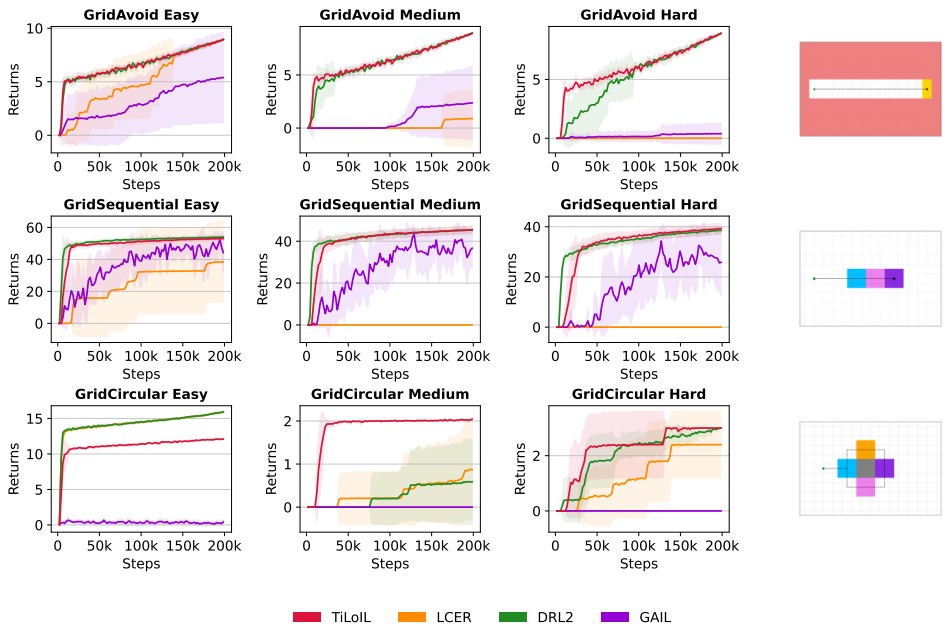

Figure 5: Discrete environment evaluation. The three rows represent reach-avoidance, sequential, and circular tasks, as shown on the right.

# E   Tabular Results

The evaluation environment is a deterministic 2D gridworld where the agent can move one unit in any of the four cardinal directions at each timestep or stay still.

Fig. 5 presents the performance of various algorithms across different GridWorld tasks under Tabular Q-Learning. The three rows correspond to distinct task categories: reach-avoidance (top row), sequential (middle row), and circular (bottom row), with increasing levels of difficulty (Easy, Medium, Hard) from left to right. The x-axis denotes the number of environment steps (up to 200k), while the y-axis represents the cumulative returns achieved.

The first row of Figure 5 shows the results in three *Grid Avoid* environments with LTL specification $\varphi = \mathsf{F}(a) \wedge \mathsf{G}\neg b$, with different grid sizes. The agent must navigate around a large area encompassing everything except a narrow corridor. The goal area is located at the end of this corridor, and the task becomes more challenging as the corridor's length increases. Here, the discriminators of our method will assign a bad reward whenever the agent exits the corridor, causing the LDBA to transition to a sink state. The discriminators direct the trajectory to align with the behavior observed in the demonstrations. Therefore, it has enhanced learning efficiency. Our approach shows a consistent and significant improvement in returns, especially in the hard setting.

The second and third row of Figure 5 shows the results in Sequential environments with LTL specification $\varphi = \mathsf{F}(a \wedge \mathsf{XF}(b \wedge \mathsf{XF}(c))$ and circular environments with **LTL** specification $\varphi = \mathsf{GF}(a \wedge \mathsf{XF}(b \wedge \mathsf{XF}(c))\mathsf{G}\neg b$. Here, a, b, and c are different zones. The agent needs to reach each zone in order. The hardness increases with the number of zones. For the sequential task, the number of zones increases progressively, with 3, 4, and 5 zones for the easy, medium, and hard levels, respectively. For the circular task, each mode has 2, 3, and 4 zones. In the first scenario, the agent must visit a specific sequence of zones in a predetermined order. Our method quickly achieves optimal returns, outperforming other methods. In the second scenario, the agent must repeat this sequence indefinitely while avoiding the center of the room. In this case, our method not only converges faster but also achieves significantly higher returns compared to other baselines, particularly in the medium and hard settings. For the *GridCircular Medium*, each trajectory has 72 steps; the *GridCircular Hard* 144 steps in each trajectory.

Table 1: Effect of the reward-shaping coefficient $\beta$ on TiLoIL performance, measured by eventual discounted returns over 10 seeds.

| Env | $\beta$=0.49 | $\beta$=1/3 | $\beta$=1/4 | LCER | DRL2 | GAIL | CYCLER |
|---|---|---|---|---|---|---|---|
| GridCircular Hard | 3±0 | 3±0 | 3±0 | 2.4±1.2 | 3±0 | 0±0 | 3±0 |
| FlatWorld Stabilization | 47±0 | 47±0 | 47±0 | 47±0 | 46.8±0.5 | 9.6±19 | 47±0 |
| FlatWorld Cycle | 1.5±1.5 | 2.3±0.8 | 1.5±0.8 | 1.6±0.6 | 0±0 | 0±0 | 0±0 |
| Doggo Avoid | 268±78 | 296±74 | 246±110 | 4.2±12.5 | 79.9±67.1 | 0±0 | 164.6±125 |
| Doggo Navigate | 342±38 | 246±61 | 192±81 | 0±0 | 45.9±64.7 | 0±0 | 0±0 |
| Fetch Avoid | 35±1.9 | 36±2.8 | 37±1.5 | 13.7±14.8 | 35.7±5.5 | 34.9±6.7 | 12.1±8.8 |
| Fetch Align | 30.1±6.4 | 21±15.9 | 20±1.8 | 1.8±3.3 | 0±0 | 0±0 | 0±0 |
| Carlo | 2.6±1.1 | 3.2±1.8 | 2.3±0.8 | 0±0 | 0.3±0.5 | 0±0 | 0±0 |
| CheetahFlip | 7.25±1.3 | 3.5±2.1 | 3.3±3.3 | 0±0 | 2.5±5 | 2.4±1.6 | 0±0 |

Table 2: Comparison between TiLoIL with multiple discriminators and a shared global discriminator, measured by eventual discounted returns (mean ± standard deviation) over 5 seeds.

| Env | TiLoIL (multiple discriminators) | TiLoIL (shared discriminator) |
|---|---|---|
| Doggo Avoid | 296±74 | 360±20 |
| Doggo Navigate | 246±61 | 286±91 |
| Fetch Avoid | 36±2.8 | 37±0.7 |
| Fetch Align | 21±15.9 | 26±22 |
| Carlo | 3.2±1.8 | 3.9±0.5 |
| CheetahFlip | 3.5±2.1 | 3.5±2.5 |
| FlatWorld Cycle | 2.3±0.8 | 2.8±0.1 |

# F    Additional Ablation Studies

## F.1    $\beta$ Sensitivity

In the reward formulation Eq. 8, $\beta$ controls how much the shaped rewards from multi-stage discriminators weigh. Table 1 shows how different values of $\beta$ : $(0.49, 1/3, 1/4)$ affect performance across environments, measured by eventual discounted returns over 10 seeds.

All the ablations outperform the baselines. For easy tasks like GridCircular Hard and FlatWorld Stabilization, this hyperparameter doesn't have a significant impact. Some challenging tasks such as Fetch Align and CheetahFlip perform better with larger $\beta$, meaning the shaped reward from the discriminator guides the learning more effectively.

## F.2    Scalability: Shared vs. Multiple Discriminators

More complex LTL specifications can increase the number of discriminators, making training more computationally intensive and potentially unstable. We have conducted additional experiments to address this scalability concern. In our reward formulation in Eq. 8, $\mathrm{SIdx}(b)$ serves as an approximation of the stage index of the product MDP state $(s, b)$, $\beta$ is a hyperparameter, and $f_b(s)$ is a discriminator associated with LDBA state $b$. To mitigate the scalability issue of training a separate discriminator per LDBA state, we experiment with a single global discriminator $f(b, s)$, which is conditioned on the current LDBA state $b$, and reformulate the reward as

$$\mathcal{R}((s, b)) = \frac{\mathrm{SIdx}(b) + \beta \tanh(f(b, s))}{\mathcal{N}(b)}.$$

In implementing $f(b, s)$, we concatenate a one-hot encoding $z_b \in \mathbb{R}^K$ of the LDBA state $b$ (where $K$ is the number of LDBA states) with the state vector $s$ as input to the discriminator network.

As in the original implementation, we train $f(b, s)$ to predict whether an MDP state $s \in S$ originates from a positive trajectory whose maximal stage exceeds the LDBA state $b$, as opposed to a negative trajectory that fails to progress beyond $b$ or gets trapped in a sink state. As shown in Table 2, the shared global discriminator achieves comparable or better performance than the per-state variant, demonstrating improved scalability without loss of effectiveness.

We hypothesize that the improvement arises from more efficient data sharing across LDBA states, which stabilizes training and mitigates overfitting to individual states with limited positive samples. This result suggests that structured conditioning of a shared discriminator is a promising future direction to improve scalability without compromising performance (for example, one could future explore introducing FiLM layers that modulate hidden activations in the shared discriminator based on the LDBA state).

### F.3 Robustness under Suboptimal Demonstrations

To examine TiLoIL's robustness under suboptimal demonstrations, we consider cases where demonstrations fail to reach LDBA accepting states. In such cases, the underlying RL component must discover these transitions through exploration. As described in Line 4 of Algorithm 1, we assign demonstration trajectories to the stage buffers corresponding to LDBA accepting states, even if they fail to fully complete the task. Demonstrations that fall short of fully satisfying the LTL formula—such as those that progress through some sub-goals or approach accepting states without fully reaching them—can still provide valuable learning signals during training.

We validate this in Carlo environment. This environment involves a self-driving agent trained to follow a circular track counterclockwise while visiting two blue regions repeatedly and avoiding crashes. We trained TiLoIL using only failed or incomplete demonstrations, including those that either entered unsafe regions or failed to complete a full loop by missing one blue region. While there is a performance drop—as expected—TiLoIL remains effective, significantly outperforming the RL for LTL baselines and GAIL, as reported in the main paper.

Table 3: Performance comparison between TiLoIL trained with original demonstrations and with only unsafe or incomplete demonstrations.

| Environment | TiLoIL (Original Demos) | TiLoIL (Unsafe or Incomplete Demos Only) |
|---|---|---|
| Carlo | 3.2±1.8 | 2.1±0.5 |

## G  Demonstrations

### G.1  Demonstration Assumption

Our primary contribution is to show that finite-length, non-optimal demonstrations—collected by suboptimal policies that reach LDBA accepting states only once or twice—can be leveraged to learn policies that satisfy LTL objectives over infinite horizons. The key lies in generalizing from such limited behaviors to policies that reach accepting states infinitely often. If demonstrations fail to cover certain transitions to LDBA accepting states, it becomes the responsibility of the underlying RL algorithm (SAC in TiLoIL) to discover trajectories that reach those states through its own exploration. TiLoIL is able to leverage the agent's own behaviors—those that have progressed beyond certain LDBA states—as positive trajectories to train discriminators that guide learning for those trajectories that fail to meet such LDBA states as in Algorithm 1, thereby reducing reliance on large demonstration datasets. However, if the RL algorithm fails to explore LDBA accepting states, TiLoIL is likely to fail. A promising future direction is to combine TiLoIL's exploitation strategy—which leverages demonstrations—with advanced exploration techniques from recent RL-for-LTL algorithms, aiming to overcome the reliance on demonstrations as a bottleneck.

### G.2  Demonstration Sizes

Demonstrations can be difficult to obtain, especially in complex environments, making it important to work with a limited number of demonstrations. In Fig. 3, the results are based on 5 demonstrations. We aim to investigate whether more demonstration data would improve learning. As shown in Fig. 6, the demonstration size does not significantly limit our learning efficiency, though it has some impact on challenging tasks like Fetch Align and Cheetah Flip. This is mainly because TiLoIL trains multi-stage discriminators from stage-specific buffers. TiLoIL is able to leverage the agent's own behaviors—those that have progressed beyond certain LDBA states—as positive trajectories to train

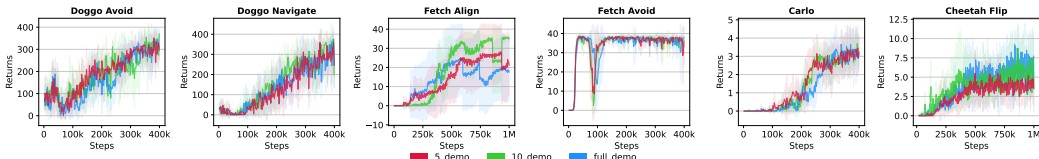

Figure 6: This figure illustrates our method with different numbers of demonstrations. The lines represent 5, 10, and full-size demonstrations. The full demonstration (number of episodes) varies across environments due to differences in episode length. Specifically, the full demonstration numbers are as follows: *Doggo Avoid* is 25, *Doggo Navigate* is 15, *Fetch Align* and *Fetch Avoid* are 100, *Carlo* is 111, and *Cheetah Flip* is 50.

discriminators that guide learning for those trajectories that fail to meet such LDBA states, thereby reducing reliance on large demonstration datasets. We conclude that our method does not require large numbers of demonstrations.

We also compared our method with GAIL across varying numbers of demonstrations. As shown in Fig. 7, while GAIL improves with more demonstrations, TiLoIL consistently outperforms it by a significant margin.

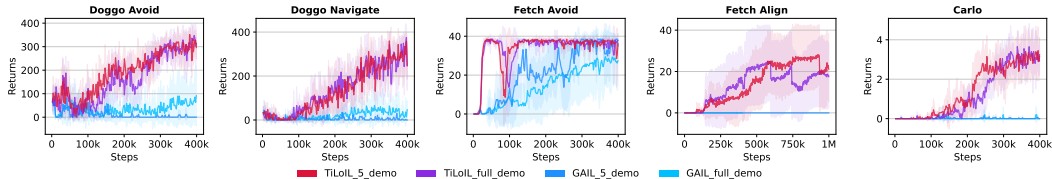

Figure 7: Returns under eventual discounting comparing TiLoIL with GAIL under different sizes of demonstrations. The full demonstration (number of episodes) varies across environments due to differences in episode length.

### G.3 Demonstration Lengths

Table 4 illustrates the impact of demonstration length on performance. Performances are measured by eventual discounted returns over 10 seeds. The first column uses demonstrations of 100 steps, while the second and third columns extend this to 200 and 400 steps, respectively. In the Carlo environment, longer demonstrations improve learning outcomes. For CheetahFlip, extending the demonstration length does not help because the demonstrations are suboptimal, and longer trajectories do not provide more visits to accepting states.

Table 4: Effect of demonstration length on TiLoIL performance, measured by eventual discounted returns over 10 seeds.

| Env | demo length=100 | 200 | 400 |
|---|---|---|---|
| Carlo | 1.6±0.7 | 2.15±0.9 | 3.2±1.8 |
| CheetahFlip | 3.5±2.1 | 3.3±2.3 | 2.3±1.8 |

## H Implementation Details

### H.1 Environments and Tasks

The experiments presented in this paper are conducted within simulated environments.

### H.1.1 Tabular Environments

The environments explored in the Fig. 5 are variations of the standard gridworld, represented as a 2D grid with the agent occupying a single cell. The action space is discrete, comprising five actions: four corresponding to movements in the cardinal directions and one no-op action. The observation space is 2-dimensional, consisting of the agent's x and y coordinates.

Although the dynamics are consistent across tasks, the labeling functions mapping MDP states to atomic propositions, as well as task-specific details, differ from each other.

**Reach-avoidance** Reach-avoidance is assessed in a gridworld without obstacles as shown in the first row of Fig.5. The agent starts at one end of a narrow corridor with a unit-width layout. While the opposite end of the corridor is unbounded, the atomic proposition $a$ evaluates to true in all cells of the corridor located beyond a certain fixed distance from the starting position. Conversely, the AP $z$ evaluates to true in all areas outside the corridor. The LTL formula is $F(a) \land G\neg z$. The task's difficulty increases with the distance to the target zone, set to 7, 9, and 11 for the easy, medium, and hard variants, respectively. Each episode lasts for the minimum number of steps required to reach the target zone, plus an additional 10 steps.

**Sequential** Sequential task is shown in the second row of Fig. 5, it also takes place in a gridworld without obstacles, consisting of contiguous 7×7 squares aligned horizontally. The agent starts at the center of the leftmost square, and in each subsequent square to the right, a different atomic proposition (AP) evaluates to true in alphabetical order. Specifically, $a$ evaluates to true in the first square to the right of the starting position, $b$ in the second, and so on. The task's difficulty is scaled by using progressively longer temporal logic formulas: $F(a \land XF(b \land XFc))$, $F(a \land XF(b \land XF(c \land XFd)))$, and $F(a \land XF(b \land XF(c \land XF(d \land XFe))))$ for the easy, medium, and hard variants, respectively. Each episode has a fixed length of 72 steps.

**Circular** Circular tasks are shown in the third row of Fig. 5 They consist of 5 contiguous 7 × 7 squares arranged in a cross formation. The central zone with 6 × 6 cells acts as an obstacle, labeled with the atomic proposition $z$, and the agent is unable to access it. The remaining 4 zones are labeled a, b, c, and d in a counterclockwise order, with the agent initialized near the first zone. Task difficulty is scaled by increasing the number of zones involved in the loop, with the following formulas used: $GF(a \land XF(b)) \land G\neg z$ for the easy variant, $GF(a \land XF(b \land XFc)) \land G\neg z$ for medium, and $GF(a \land XF(b \land XF(c \land XFd))) \land G\neg z$ for hard. Each episode has 72 steps for easy and medium mode; hard mode has 144 steps.

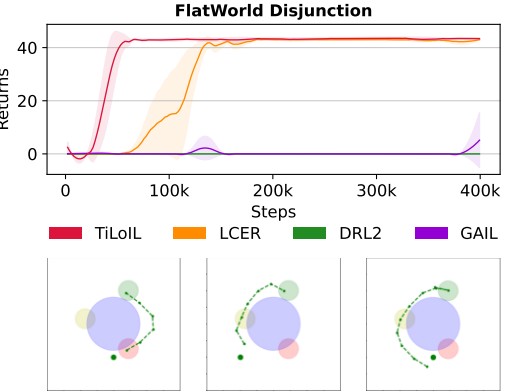

Figure 8: Disjunctive LTL objective in FlatWorld. The task requires reaching (yellow → green) or (red → green) while avoiding blue. The first row shows that returns under eventual discounting comparing TiLoIL with the baselines over 10 random seeds. The second row illustrates the trajectories of TiLoIL and LCER. The first image shows TiLoIL starting near the red zone, while the second shows TiLoIL starting near the green zone. The third image depicts the LCER trajectory from a red-zone start. Although LCER achieves similar returns to TiLoIL, it fails to adapt its path based on the initial state, always following yellow → green. In contrast, TiLoIL dynamically selects the optimal path (e.g., red → green when starting closer to red).

### H.1.2 Continuous Environments

**Carlo** The Carlo environment (illustrated in the second row, last column of Fig. 3) is a simplified self-driving simulator based on a bicycle model for its dynamics. The agent observes its position, velocity, and heading (in radians), resulting in a 5-dimensional observation space. The agent controls its heading and throttle, with an action space of $[-1, 1]^2$. For this domain, we use a circular track where the agent starts at the center of the road at an angle of $\frac{(1+2i)\pi}{4}$, $i = 0$ and drives counterclockwise around the circle without crashing. The task is defined by $GF(wp_0 \land XF(wp_1)) \land G\neg crash$, to visit the blue regions $wp_0, wp_1$ repeatedly while avoiding the gray region $crash$. The episode length is 500.

**Doggo** Doggo is a 12-DoF quadruped adapted from the most challenging tasks in SafetyGym [52], designed to navigate a flat plane. The observation space is 66-dimensional, the action space is 12-dimensional, and each episode lasts 500 steps. Similar to other reach-avoidance tasks, *Doggo Avoid* requires the agent to navigate directly to a distant goal along a straight path, avoiding any detours as shown in the first column, the second row of Fig. 3. In contrast, *Doggo Navigate* involves navigating through a sequence of two circular zones as shown in the first column, the third row of Fig. 3. These tasks are defined by the following specifications: $\mathsf{F}a \wedge \mathsf{G}\neg z$ for *Doggo Avoid* and $\mathsf{F}(a \wedge \mathsf{X}\mathsf{F}(b))$ for *Doggo Navigate*. Both tasks have a fixed episode length of 500 steps.

**FlatWorld** The Flatworld environment (illustrated in the first row of Fig. 3) is a two-dimensional continuous world. The agent, represented by a green dot, starts at position $(-1, -1)$. The dynamics of the environment are defined by: $x = x + \frac{a}{10}$ where $x \in \mathbb{R}^2$ and $a \in [0, 1]^2$. Here we define three tasks *FlatWorld Stabilization*, *FlatWorld Cycle* and *FlatWorld Disjunction*. *FlatWorld Stabilization* also inspired by the reach-avoidance task in tabular settings, requires the agent to reach the yellow zone at left and avoid the blue zones in the middle. *FlatWorld Cycle* requires the agent to visit red, yellow, and green zones in order repeatedly while avoid the blue zone. *FlatWorld Disjunction* requires the agent to visit yellow then green, or red then green while avoid the blue zone. The three tasks are defined by the following specifications: $\mathsf{F}\mathsf{G}(y) \wedge \mathsf{G}\neg b$ for *FlatWorld Stabilization*, $\mathsf{G}\mathsf{F}(r \wedge \mathsf{X}\mathsf{F}(g \wedge \mathsf{X}\mathsf{F}y)) \wedge \mathsf{G}\neg b$ for *FlatWorld Cycle*, and $(\mathsf{F}(y \wedge \mathsf{X}\mathsf{F}g) \vee \mathsf{F}(r \wedge \mathsf{X}\mathsf{F}g)) \wedge \mathsf{G}\neg b$ for *FlatWorld Disjunction*. Fig. 8 shows the result for *FlatWorld Disjunction*. The episode length is 50 for all tasks.

**Fetch** Fetch environments are based on the widely used Fetch robotic benchmark [17]. We evaluate two tasks where the agent controls the end effector position of a 7-DoF robotic arm using 4-dimensional actions over 50-step episodes. *Fetch Avoid*, inspired by the reach-avoidance task in tabular settings, requires the arm to fully extend while avoiding lateral movements (i.e., reaching the green zone and avoiding red zones as shown in the second row of Fig. 3). In this task, the observation space is 10-dimensional. *Fetch Align*, on the other hand, involves interacting with three cubes on one side of the table and aligning them horizontally at the center. For this task, the observation space includes information about the cubes and is 45-dimensional. The two tasks are defined by the following specifications: $\mathsf{F}a \wedge \mathsf{G}\neg z$ (reach the end of the table while avoiding lateral movements) for *Fetch Avoid*, and $\mathsf{F}(a \wedge \mathsf{X}\mathsf{F}(b \wedge \mathsf{X}\mathsf{F}c))$ (position the first, second, and third block sequentially) for *Fetch Align*.

**FetchPlace** FetchPlace environments are based on the widely used Fetch robotic benchmark [17]. We evaluate two tasks where the agent controls the end effector position of a 7-DoF robotic arm using 4-dimensional actions over 500-step episodes. In *FetchPlace Button*, pressing the button causes a goal region to appear. The manipulator must (1) press the button, (2) pick up a block, and move it to the goal region, and then (3) repeat this sequence with the goal region appearing in a new location each time. In this task, the observation space includes the button information and goal information, and is 31-dimensional. *FetchPlace Tray* contains a movable block and two potential destination regions: a tray and a goal region. A tray may or may not appear in the scene in each episode. If the tray appears, the agent must place the block into the tray. If the tray is absent, the block should be placed into the goal region. For this task, the observation space includes information about the goal and is 28-dimensional. The two tasks are defined by the following specifications: $\mathsf{F}grasp \wedge \mathsf{G}\mathsf{F}(GoalReached \wedge \mathsf{X}\mathsf{F}ButtonReached$ (first grasp the block, reach the goal, and then reach the button) for *FetchPlace Button*, and $(\neg TrayPresent \vee \mathsf{F}(grasp \wedge \mathsf{X}\mathsf{F}InTray)) \wedge (TrayPresent \vee \mathsf{F}(grasp \wedge \mathsf{X}\mathsf{F}InGoal))$ (if tray not present, grasp the block and go into the tray; if tray present, grasp the block and go to the goal) for *FetchPlace Tray*.

**HalfCheetah** HalfCheetah [61], a standard environment in deep reinforcement learning, involves controlling a 6-DoF robot in a vertical 2D plane. In *Cheetah Flip*, the agent follows the formula $\mathsf{G}\mathsf{F}(b \wedge \mathsf{X}\mathsf{F}d)$, where each variable corresponds to a specific range of angles for the robot's main body. These angles represent the Cheetah standing on its front legs and standing on its back legs, respectively. The task requires the agent to perform a sequence of frontflips to satisfy the formula. The episode length is 1000.

**SixteenRooms** This environment with continuous observation and action space is adapted from the 16 rooms environment [35]. In this walled environment with 16 rooms, each room has the same size 8×8 divided by walls and corridors with thickness. We deployed the FlatWorld agent in this environment. The task $\Psi$ requires the agent to traverse a circular path indefinitely via one of two

Table 5: Hyperparameters for Q-learning and SoftActorCritic.

| HYPERPARAMETER | VALUE |
|---|---|
| $\gamma$ | 0.99 |
| $\alpha$ | 0.2 |
| BUFFER SIZE | $1 \cdot 10^6$ |
| BATCH SIZE | 64 |
| LEANING STARTS | 2000 |
| $\tau$ | $1 \cdot 10^{-4}$ |
| Q LEARNING RATE | $3 \cdot 10^{-4}$ |
| ACTOR LEARNING RATE | $3 \cdot 10^{-4}$ |
| CRITIC LEARNING RATE | $3 \cdot 10^{-4}$ |
| DISCRIMINATOR LEARNING RATE | $3 \cdot 10^{-4}$ |
| DISCRIMINATOR UPDATE FREQUENCY | 1 STEP |
| TARGET NETWORK UPDATE FREQUENCY | 1 STEP |

possible routes. The rooms are arranged in a $4 \times 4$ grid, and are indexed as $\texttt{room}_{i,j}$ with $i$ denoting the row index (from left to right) and $j$ the column index (from bottom to top).

The LTL formula: $\Psi = \Big( \mathbf{GF}(\texttt{room}_{1,2} \wedge \mathbf{XF}(\texttt{room}_{1,4} \wedge \mathbf{XF}(\texttt{room}_{3,4} \wedge \mathbf{XF}(\texttt{room}_{3,2})))) \vee \mathbf{GF}(\texttt{room}_{3,1} \wedge \mathbf{XF}(\texttt{room}_{3,3} \wedge \mathbf{XF}(\texttt{room}_{3,4} \wedge \mathbf{XF}(\texttt{room}_{4,4} \wedge \mathbf{XF}(\texttt{room}_{4,3})))))) \Big) \wedge \mathbf{G}(\neg\texttt{Wall})$ specifies that the agent must repeatedly visit either one of two possible loops. The large loop goes through rooms $(1,2) \to (1,4) \to (3,4) \to (3,2)$. The small loop goes through rooms $(3,3) \to (3,4) \to (4,4) \to (4,3)$. The agent must avoid collisions with wall segments between rooms. The agent departs from the center of the bottom-left room to reach the desired goal positions. The initial state of the agent is created with random noise. The optimal policy needs to identify and traverse the smaller loop repeatedly while avoiding all walls.

## H.2 Methods and Algorithms

We utilize SAC [25] and tabular Q learning [66] as the backbone RL algorithm. The architectures of the SAC networks are shown below:

- Actor Network:4-layer MLP, hidden units(256,256,256)
- Critic Networks:3-layer MLP, hidden units(256,256)
- Discriminator Networks(Reward):2-layer MLP,hidden units(32)

The corresponding hyperparameters are provided in Table 5. For challenging tasks, we allow more steps for initial random exploration. For *Cheetah Flip*, learning starts is $2 \cdot 10^4$.

## H.3 Metrics

Performance is evaluated using cumulative return under eventual discounting [63], an RL-friendly proxy objective from prior work [27, 63] that optimizes a lower bound on the probability of LTL formula satisfaction.

$$\pi_\gamma^\star \in \arg\max_{\pi \in \Pi} \mathbb{E}_{\tau \sim \pi}\left[ \sum_{i=0}^{\infty} \Gamma_i R^\times(b_i) \right] (=: V_\pi^\gamma),$$

where

$$R^\times(b_i) = 1_{\{b_i \in \mathcal{B}^*\}}, \quad \Gamma_0 = 1, \quad \Gamma_i = \prod_{t=0}^{i-1} \gamma^\times(b_t), \tag{17}$$

and

$$\gamma^\times(b_t) = \begin{cases} \gamma, & b_t \in \mathcal{B}^*, \\ 1, & \text{otherwise.} \end{cases} \tag{18}$$

All performance curves represent the mean estimated across 10 seeds, with shaded areas indicating variance.

### H.4 Tools

Our codebase primarily utilizes NumPy [26] for numerical computations and Torch [49] for its autograd capabilities. Additionally, we partially automate the synthesis of LDBAs from LTL formulas using Rabinizer [40].

### H.5 Computational Costs

All methods exhibit similar runtimes within their respective settings (tabular or continuous). Each experimental run was conducted using NVIDIA Quadro RTX 6000 GPU. On average, a single run took approximately 30 minutes in the tabular setting and 3 hours in the continuous setting.

