# OpenReview forum: "Imitation Learning with Temporal Logic Constraints"
_NeurIPS.cc/2025/Conference — NeurIPS 2025 poster_

### Official Review · Reviewer_Tszg · 2025-06-14

**Clarity:** 4
**Significance:** 2
**Originality:** 3
**Rating:** 5
**Confidence:** 4

**Summary:**

In this paper, the authors address the problem of learning to accomplish infinite-horizon tasks posed as LTL formulas. In these tasks, the goal is to repeatedly reach an accepting state in the Limit Deterministic Büchi Automaton (LDBA) that results from the LTL. These tasks are challenging since they have naturally sparse rewards. To address this, the authors propose to leverage segmented finite expert demonstrations and trajectories produced by the learned policy itself. The method enables providing guidance for each segment and results in faster convergence with respect to several baselines.

**Questions:**

I would really appreciate if the authors can especially focus on Q1 and Q2. I am keen on increasing my score if my doubts are addressed.

**Q1.** The approach seems general enough to be applicable in the context of a finite-horizon task since finite-horizon tasks can also be naturally segmented. Indeed, most of the tasks used in the ablations (`Doggo` and `Fetch`) are finite-horizon in my understanding, since they have absorbing accepting states. However, upon testing the segmentation in the ablation studies (Fig. 4) it is said that there is negligible difference between segmenting and not doing so. My questions on this matter are:
* Why would segmentation only benefit tasks without absorbing accepting states?
* Following lines 187-188, does this have something to do with satisfying LTL constraints over infinite horizons being more challenging? Why?
* Why motivating the use of infinite-horizon tasks but then most of the tasks are finite-horizon?
* Why are the ablations not performed (or shown) for `GridCircular Hard` or `FlatWorld` tasks?
Please let me know I am wrong in saying that tasks with absorbing accepting states do not have an infinite horizon.

**Q2.** In what contexts would finite-length demonstrations (or a finite number of them) be potentially problematic or insufficient? In the considered tasks, the environment never changes after reaching the accepting state (e.g. the yellow area in `FlatWorld Stabilization` does not change location); thus, a single demonstration where the accepting state is reached once seems enough. However, there are infinite-horizon tasks where the environment can change, e.g. see the tasks Cookie (the location of a cookie changes after pressing a button) and Symbol in [1]. Under which circumstances would demonstrations in these settings be useful? Have any inherent assumptions on the demonstrations been made in the submitted work? Is the assumption visiting an accepting state only once or twice stronger than it seems?

**Q3.** Has the effect of having longer or shorter demonstrations (hence, reaching the accepting states more or fewer times) been studied? If not, how do you think it can influence the performance of the algorithm?

**Q4.** Has the approach been compared against methods that leverage the structure of an automaton for reward shaping? For example, potential-based reward shaping is employed in the cited works by Camacho et al. using the minimum distance to the accepting state and performing value iteration over the automaton graph. How about method using hierarchical reinforcement learning, which similarly leverage the decomposition of a task into substages? If not, what are the key differential factors you would expect to see (or if they are not applicable in your context, please explain why).

**References**
[1] Toro Icarte et al. (2019). Learning Reward Machines for Partially Observable Reinforcement Learning.

**Ethical Concerns:**

["NO or VERY MINOR ethics concerns only"]

**Final Justification:**

In light of the positive answers provided by the authors, I have decided to increase my score. The paper explores a novel direction at the intersection between RL and LTL, focusing on learning policies for infinite-horizon tasks from finite-length imperfect demonstrations. The motivation is clear, the core contributions are well described, and several ablations are performed. Key work in the area---at least the ones I am aware of---are acknowledged.

My concerns about the method were mainly on the experimental side. Most domains did not seem to be infinite-horizon, but they actually were, despite using absorbing states. Reservations about implicit assumptions on the limitations have also been addressed through domains recommended in the review. Research in the area would benefit from adding them in future research.

In the discussion with the authors, [some recommendations](https://openreview.net/forum?id=MKLcSmSTQI&noteId=n2J8BZYPNQ) were made for future versions. The authors also [listed](https://openreview.net/forum?id=MKLcSmSTQI&noteId=k6zvvUT3cr) the main "to-do" for the following versions, which will strengthen the paper.

In addition to these, the authors thoroughly addressed the concerns of other reviewers. For future versions, I would especially highlight:
- Some of the suggested related work and the use of the term "multi-objective" by Reviewer ifPj.
- Following the discussion with Reviewer uyFt, the authors could point out some existing approaches that use rewards encoding _progress toward specification satisfaction_, e.g. the LTL-based works [1, 2], which give a reward of 1 every time the LTL formula is progressed.
- Assumptions (or lack thereof) in the demonstrations, following Q3 answer to Reviewer uyFt.

[1] Vaezipoor et al. "LTL2Action: Generalizing LTL Instructions for Multi-Task RL". ICML 2021.  See Definition 3.2.

[2] Tuli et al. "Learning to Follow Instructions in Text-Based Games". NeurIPS 2022. See Section 4.2.

**Limitations:**

Yes

**Quality:**

3

**Strengths And Weaknesses:**

## Strengths
- The motivation and the background are clear and described with the right amount of detail. The running example is of great help throughout the paper.
- The core contributions (segmentation and discriminating trajectories across multiple stages) are clearly described. The ablation studies assess their impact in different tasks, and clear performance gains are shown.
- Excellent scholarship -- key works in the area are acknowledged throughout the paper.
- The assumptions and limitations of the method are clearly highlighted.

## Weaknesses
- The term *occupancy measures* in line 169 is not defined then. Following a (presumably) new mention in line 349, I understand it is related to task satisfaction. It would be convenient to clarify it in future versions.
- Figure 5 is key for the paper, but it is in the appendix.
- The figure for `FlatWorldStabilization` would be clearer if the agent stayed within the yellow zone (for clarity, seems stabilizing seems a bit ambiguous).
- The emphasis of the paper is on infinite-horizon tasks, but most of the testing tasks (`Doggo` and `Fetch`, as highlighted in Fig. 4) are (or seem) finite-horizon. The motivation upon reaching this prt of the paper becomes unclear [see Question 1].
- It would be interesting to see how performance changes with demonstrations where an accepting state is reached a variable number of times and highlight potential assumptions on the demonstrations as well as their applicability in contexts where the environment can change [see Questions 2 and 3].

## Minor Comments
- Line 124: "incentivize" -> "incentivizes".
- Line 210: "data sampled from a sampled replay buffer" -> Perhaps the authors meant "data sampled from a replay buffer".
- MaxStage in Eq. 7 -> \textsc{MaxStage} (like in the lines below).
- Line 328: footnote should go after the dot, i.e. "...stays still.\footnote{...}".
- The Conclusion should be in its main section (probably left within the Related Work because of lack of space).

---

> ### Author Rebuttal · Authors · 2025-07-29
>
> We appreciate your insightful feedback and constructive comments!
>
> **Q1.1 Why would segmentation only benefit tasks without absorbing accepting states?**
>
> In tasks with absorbing accepting states (e.g., Doggo, Fetch), the agent is expected to remain in the accepting state once it is reached, leaving no further progression to segment or imitate—aside from maintaining safety (e.g., avoiding sink states). In such cases, segmentation reduces to a trivial one-step segment exactly at accepting states and primarily serves to provide safety-related reward signals. Segmentation is particularly beneficial for tasks without absorbing accepting states, where the agent must repeatedly visit, leave, and return to accepting states. These tasks exhibit rich, multi-phase behaviors, and segmented imitation enables TiLoIL to learn policies that cycle between accepting states—effectively supporting repeated satisfaction of the LTL objective. **We emphasize that even in tasks with absorbing accepting states, TiLoIL still benefits from multi-stage discriminator learning and reward formulation, as shown in Figure 4.**
>
> **Q1.2 Following lines 187-188, does this have something to do with satisfying LTL constraints over infinite horizons being more challenging? Why?**
>
> Lines 187–188 highlight the challenge of imitation learning under LTL constraints: the agent must generalize from *finite-length* demonstrations to satisfying *infinite-horizon* objectives. In practice, user-provided demonstrations may be suboptimal due to the cost and complexity of constructing trajectories that robustly satisfy such objectives. In our setup, demonstrations are generated by chaining locally trained controllers for each stage of the LTL task (line 341). These policies are not globally optimal: when executed repeatedly, they often fail to produce trajectories that visit LDBA-accepting states infinitely often. For example, in CheetahFlip, the chained controllers—and thus the resulting demonstrations—achieve only one or two rotations, since additional flips require maintaining momentum after reaching an accepting state, which the learned local policies cannot sustain. Similarly, in Carlo, demonstrations rarely reach accepting states more than twice. Despite this suboptimality, TiLoIL learns policies that outperform the demonstrations, empirically visiting accepting states more frequently in both CheetahFlip and Carlo.
>
> **Q1.3 Why motivating the use of infinite-horizon tasks but then most of the tasks are finite-horizon? (Please let me know I am wrong in saying that tasks with absorbing accepting states do not have an infinite horizon.)**
>
> While several tasks in our benchmark suite include absorbing accepting states, they are still formally infinite-horizon: the agent operates in an unbounded setting where episodes do not terminate upon reaching accepting states. Crucially, the agent must continue to satisfy any ongoing safety properties—e.g., avoiding unsafe regions in Doggo Avoid—even after accepting states are reached. Importantly, 5 out of 9 tasks in our suite require policies with meaningful $\omega$-regular structure to satisfy the specification—for example, visiting certain regions repeatedly. These tasks truly exhibit the types of behaviors that infinite-horizon LTL specifications are designed to express. We will clarify the distinction between infinite-horizon formulations and absorbing structures in the revised version.
>
> **Q1.4 Why are the ablations not performed (or shown) for GridCircular Hard or FlatWorld tasks?**
>
> The ablations for GridCircular Hard and FlatWorld are shown below. Both multi-stage reward formulation and segmented imitation are important. Performances are measured by eventual discounted returns over 10 seeds.
>
> |Env|TiLoIL|TiLoIL-noStage|TiLoIL-noSegment|
> |-|-|-|-|
> |FlatWorld Stabilization|47±0|27.5±22.5|None|
> |GridCircular Hard|3.0±0|0±0|0±0|
> |FlatWorld Cycle|2.3±0.8|0±0|0±0|
>
> **Q2. In what contexts would finite-length demonstrations be potentially problematic or insufficient? There are infinite-horizon tasks where the environment can change, e.g. see the tasks Cookie and Symbol in [1]. Under which circumstances would demonstrations in these settings be useful? Have any inherent assumptions on the demonstrations been made in the submitted work? Is the assumption visiting an accepting state only once or twice stronger than it seems?**
>
> A finite number of demonstrations may become insufficient in more dynamic settings, such as those with varying accepting conditions, if they fail to capture the necessary behavioral diversity.
>
> To evaluate TiLoIL in such settings, we adapted two environments inspired by Cookie and Symbol from [1].
>
> * **Cookie-like task**: We modified the Fetch Pick\&Place environment by introducing a button on the table. Pressing the button causes a goal region to appear. The manipulator must (1) press the button, (2) pick up a block and move it to the goal region, and then (3) repeat this sequence with the goal region appearing in a new location each time. Compared to the baselines, TiLoIL was the only method that successfully solved this task within 2 million steps, demonstrating its ability to handle changing environments.
>
> * **Symbol-like task**: We redesigned the original Symbol task into the Pick\&Place environment. The workspace contains a movable block and two potential destination regions: a *tray* and a goal region. A tray may or may not appear in the scene at each episode. The symbolic cue is the presence or absence of the tray:
>
>     * If the tray appears, the agent must place the block into the tray.
>     * If the tray is absent, the block should be placed into the goal region.
>
> The LTL specification is (tray_present -> F in_tray) && (Not tray_present -> F in\_goal). This task highlights a core challenge in learning from partial demonstrations. When demonstrations only show placing the block into the goal region (i.e., when the tray is absent), TiLoIL fails to generalize to the unseen tray-present case. It always moves the block to the goal region regardless of tray appearance. However, when demonstrations cover both cases (tray present and tray absent), TiLoIL successfully learns the correct conditional behavior.
>
> These experiments clarify our assumption: demonstrations must sufficiently cover the behavioral variations necessary to reach the accepting conditions under different modes. When demonstrations cover only a subset of the possible cases, the learned policy may become biased toward that subset and fail to generalize. We will include a clearer articulation of this assumption, along with a discussion of these experiments, in the revised manuscript. We will clarify that “reaching accepting states once or twice” is not a criterion for demonstration quality.  Rather, this was a practical design choice in our experimental setup to showcase that TiLoIL can still succeed even when demonstrations are sparse *for the benchmarks we considered*.
>
> **Q3. Has the effect of having longer or shorter demonstrations (hence, reaching the accepting states more or fewer times) been studied? If not, how do you think it can influence the performance of the algorithm?**
>
> The table below illustrates the impact of demonstration length on performance. Performances are measured by eventual discounted returns over 10 seeds. The first column uses demonstrations of 100 steps, while the second and third columns extend this to 200 and 400 steps, respectively. In the Carlo environment, longer demonstrations improve learning outcomes. For CheetahFlip, extending the demonstration length does not help because the demonstrations are suboptimal (see Q1.2) and longer trajectories do not provide more visits to accepting states.
>
> |Env|demo length=100|200|400|
> |-|-|-|-|
> |Carlo|1.6±0.7|2.15±0.9|3.2±1.8|
> |CheetahFlip|3.5±2.1|3.3±2.3|2.3±1.8|
>
> **Q4.1 Has the approach been compared against methods that leverage the structure of an automaton for reward shaping?**
>
> Indeed, earlier works by Camacho et al. [12, 13] propose potential-based reward shaping using distances in product MDPs (based on the minimum number of steps to accepting automaton states). Another work [31] extend this idea by computing potential functions via value iteration on a synthesized MDP from the automaton of a temporal property. These approaches are not directly applicable to ω-regular problems. By adopting an LDBA instead of a DFA, and integrating an eventual discounting scheme, more recent approaches introduce reward shaping methods tailored for the full expressiveness of LTL. For example, Bagatella et al. [6] (DRL2) propose a Bayesian method that models LDBAs as Markov reward processes with learned transition kernels, enabling potential-based shaping via high-level value estimation. Similarly, Shah et al. [55] (Cycler) distribute heuristic rewards along LDBA accepting paths. In our paper, we compare TiLoIL against DRL2 and Cycler as the most relevant potential-based reward shaping baselines.  As shown in Fig. 3, TiLoIL consistently outperforms both, demonstrating the benefit of leveraging demonstration guidance.
>
> **Q4.2  How about method using hierarchical reinforcement learning, which similarly leverage the decomposition of a task into substages?**
>
> Indeed, several works within the LTL literature leverage the structure of automata to define hierarchical policies, such as Hasanbeig et al. (2020), Icarte et al. (2022), and Jothimurugan et al. (2021) [28, 31, 35]. These methods decompose the overall task into subtasks associated with automaton transitions, improving sample efficiency. However, a known limitation of such hierarchical approaches is potential suboptimality due to local policy decisions at each subtask level [31].

---

> ### Comment · Reviewer_Tszg · 2025-08-03
>
> I appreciate the authors' responses, which have positively addressed all my concerns; hence, I will likely increase my score just before the end of the discussion period.
>
> In summary, the paper would highly benefit from the following in future versions:
> - Clarifying the distinction that tasks with absorbing accepting states are still infinite-horizon (related to Q1.1-Q1.3).
> - Including the ablations in Q1.4 (it could be in the appendix if there is no space).
> - Including the adapted environments from Q2, Cookie-like task, and Symbol-like task, adds insight into the potential and the limitations of the approach. Including them in the main paper (or adding them to the appendix, with summarized findings in the main paper) would really help drive future research in the field. Adding the related _assumption_ clarification would be of great value.
> - The analysis on the demonstration length (Q3) is also interesting because it shows that longer demonstrations do not always improve performance.
> - Restructure the closing of the paper (conclusions and future work should not be within the related work).
>
> By reading the authors' answers to other reviewers, the authors should be clearer about the contribution (i.e., learning from finite-length non-optimal demonstrations), the challenges in the scenario they tackle (e.g., not necessarily optimal demonstrations), how it relates to previous work (e.g., inverse RL, which may help emphasize the possible absence of optimal demonstrations), and the assumptions on the demonstrations (also tackled in the response to this review).

---

> > ### Author Response · Authors · 2025-08-05
> > **Thank you for your feedback!**
> >
> > Thank you very much for your thoughtful and constructive feedback. We truly appreciate your positive assessment of our responses and your openness to revisiting the score.
> >
> > We fully agree with your suggestions for improving the clarity and presentation of the paper. In particular, we will revise the final version to:
> > * (1) expand the discussion in Line 133, Section 2, to clarify the assumptions TiLoIL makes about demonstrations;
> > * (2) clarify the infinite-horizon nature of tasks with absorbing accepting states in the experiment section;
> > * (3) include the requested ablations on the GridCircular Hard and FlatWorld tasks in the main paper;
> > * (4) incorporate the Cookie-like and Symbol-like tasks (based on the Fetch Pick\&Place environment) into the main paper, as they offer key insights into the strengths and limitations of TiLoIL, and use them to further clarify our demonstration assumptions;
> > * (5) add ablation results on the impact of longer demonstrations and explain when and why they may not improve performance; and
> > * (6) restructure the conclusion and future work for better clarity.
> >
> > In particular, we will emphasize that TiLoIL's primary contribution is to show that finite-length, non-optimal demonstrations—collected by suboptimal policies that reach LDBA accepting states only once or twice—can be leveraged to learn policies that satisfy LTL objectives over infinite horizons. The key lies in generalizing from such limited behaviors to policies that reach accepting states infinitely often. We will also strengthen the related work discussion as promised.
> >
> > Your comments have been instrumental in helping us refine both the framing and technical presentation of the paper. Thank you again for your engagement!

---

### Official Review · Reviewer_uyFt · 2025-06-21

**Clarity:** 2
**Significance:** 2
**Originality:** 3
**Rating:** 2
**Confidence:** 4

**Summary:**

This paper tackles the challenge of satisfying LTL objectives in reinforcement learning over infinite horizons by using finite-length demonstrations to guide exploration. The proposed method segments trajectories at accepting states and trains the agent to reach these states from arbitrary points, improving satisfaction probability without requiring long demonstrations. Experiments in high-dimensional continuous control domains show faster convergence and better performance than baseline methods.

**Questions:**

1. In Eq. (4), the reward is defined as a function of the product state $R((s,b))$, independent of the action $a$. This differs from the standard RL setting, where the reward typically depends on both the state and the action, i.e., $R(s, a)$. Could the authors clarify the rationale behind this formulation and its implications, such as how it may affect the optimality of the learned policy in cases where different actions lead to different outcomes despite receiving the same reward?
2. For the multi-stage reward formulation, how does the choice of the hyperparameter $\beta$ affect performance, given that the reward directly influences policy learning? Additionally, since the reward depends on the length of the longest acyclic path, could the authors provide more details on how this is computed and how it impacts learning, especially in cases where the path is long, which may lead to very small or vanishing reward signals?
3. The proposed method assumes access to expert demonstrations. Could the authors clarify the underlying assumptions about these demonstrations? Specifically, do the demonstrations contain trajectories that visit each accepting state, and do they provide sufficient coverage of the state space? For instance, in the FlatWorld Cycle environment, if the demonstrations only reach the yellow region, how would the method support learning policies that reach the green region? Similarly, if all demonstrations originate from the left side of the environment, would this limit the agent’s ability to explore or learn effectively when initialized from the right side?
4. In the experiments, how many episodes are averaged per evaluation (in addition to the 10 random seeds) for each environment? Also, aside from Cheetah Flip, are the initial positions of the agents and the placement of regions or objects randomly generated during evaluation?

**Ethical Concerns:**

["NO or VERY MINOR ethics concerns only"]

**Final Justification:**

I appreciate the authors' clarifications and additional experiments. However, I remain concerned about the scalability of the proposed method with respect to the complexity of LTL specifications, particularly those that induce a large number of LDBA states and multiple accepting states. While the provided results are interesting, they are still limited to relatively simple specifications, and it remains unclear how the method would perform on more complex LTL tasks. The alphabet over the atomic propositions is $2^{AP}$, meaning that LTL specifications can induce LDBAs with a large number of (key) states (e.g., significantly more than the 4 in the authors’ examples) and multiple accepting states (e.g., also significantly more than 2 in their examples). As the authors note, their method requires demonstrations that cover key LDBA transitions, particularly those involving visits to accepting states at least once or twice. However, the minimum number of required demonstrations can scale exponentially with the complexity of the specification, which is intractable to obtain in practice. Besides, when this assumption does not hold, i.e., not all key transitions are covered, introducing additional techniques to encourage the agent to actively explore the uncovered key-transition states would significantly improve the completeness of the proposed method rather than replying on "the underlying RL algorithm (SAC in TiLoIL) to discover trajectories that reach those states through its own exploration" as the authors claimed. Moreover, the experiments in the paper are oversimplified, with fixed environment configurations. This partly explains why demonstrations provide substantial benefit. However, such a setup significantly undermines the persuasiveness of the results and the demonstrated effectiveness of the proposed method.

**Limitations:**

The proposed method faces scalability challenges due to its multi-stage discriminator learning framework. Specifically, the number of accepting states ($b_N$ in Section 3.2) grows with the complexity of the LTL specification and the number of atomic propositions, leading to a corresponding increase in the number of discriminators. This scaling makes the training process more computationally demanding. Moreover, jointly training multiple discriminators alongside the Q-function and policy introduces additional optimization difficulties and may hinder stable convergence.

**Paper Formatting Concerns:**

No major formatting issues are noted.

**Quality:**

2

**Strengths And Weaknesses:**

The motivation to accelerate policy learning under LTL constraints is clear and well-grounded. The authors present a theoretical analysis of the proposed method and support it with empirical results across diverse tasks and environments. However, key aspects of the design choices and evaluation remain insufficiently justified or explored, as outlined in the questions and limitations below.

---

> ### Author Rebuttal · Authors · 2025-07-29
>
> We appreciate your insightful feedback and constructive comments!
>
> **Limitation: The proposed method faces scalability challenges due to its multi-stage discriminator learning framework. Specifically, the number of accepting states grows with the complexity of the LTL specification and the number of atomic propositions, leading to a corresponding increase in the number of discriminators. This scaling makes the training process more computationally demanding. Moreover, jointly training multiple discriminators alongside the Q-function and policy introduces additional optimization difficulties and may hinder stable convergence.**
>
> LDBAs are commonly used for LTL-to-automaton translation because they yield automata significantly smaller than deterministic Rabin or full deterministic automata, keeping the automaton size manageable while preserving the structure needed for verification and learning. Particularly, the number of accepting states in an LDBA is small—often just one in our benchmarks—and does not scale with specification complexity. In practice, all benchmark LTL specifications in our work (and related studies) produce LDBAs with ≤ 10 states, resulting in only a small number of discriminators and computationally feasible training. We acknowledge that jointly training multiple components can raise stability concerns; however, TiLoIL exhibited stable convergence in all experiments (see Figs. 3–4) with no instability beyond what is typical in adversarial imitation learning (e.g., GAIL). Our framework can be extended to share parameters across discriminators or use belief-weighted mixtures, which would reduce the number of separate networks needed.
>
> **Q1. In Eq. (4), the reward is defined as a function of the product state $R((s,b))$, independent of the action $a$. This differs from the standard RL setting, where the reward typically depends on both the state and the action, i.e., $R(s, a)$. Could the authors clarify the rationale behind this formulation and its implications?**
>
> Our formulation follows standard practice in automata-guided RL and formal methods, where the objective is to find $\pi^\star \in \arg\max_{\pi\in\Pi} P(\pi\models\varphi)$, the policy that maximizes the probability of satisfying an LTL specification $\varphi$. This probability can be written as the expected value of an indicator over trajectories satisfying $\varphi$ (Line 120). Therefore, the rewards need only encode *progress toward specification satisfaction* (i.e. reaching an LDBA-accepting state), which is determined solely by the product state $(s,b)$ and not by the specific action taken. On the other hand, although $R((s,b))$ does not directly distinguish between actions, the transition dynamics do: different actions from the same state lead to different future product states, resulting in different expected cumulative rewards. Consequently, the optimal policy remains well-defined, and action choices still affect performance through their impact on future specification progress.
>
> **Q2.1 For the multi-stage reward formulation, how does the choice of the hyperparameter $\beta$ affect performance, given that the reward directly influences policy learning?**
>
> In our reward formulation, $\beta$ controls how much the shaped rewards from multi-stage discriminators weigh. All the experiments in the paper are done by $\beta = 1/3$, the table below shows how different values of $\beta: (0.49, 1/3, 1/4)$ affect performance across environments, measured by eventual discounted returns over 10 seeds.
>
> |Env|Beta=0.49|Beta=1/3|Beta=1/4|LCER|DRL2|GAIL|CYCLER|
> |-|-|-|-|-|-|-|-|
> |GridCircular Hard|3±0|3±0|3±0|2.4±1.2|3±0|0±0|3±0|
> |FlatWorld Stabilization|47±0|47±0|47±0|47±0|46.8±0.5|9.6±19|47±0|
> |FlatWorld Cycle|1.5±1.5|2.3±0.8|1.5±0.8|1.6±0.6|0±0|0±0|0±0|
> |Doggo Avoid|268±78|296±74|246±110|4.2±12.5|79.9±67.1|0±0|164.6±125|
> |Doggo Navigate|342±38|246±61|192±28|0±0|45.9±64.7|0±0|0±0|
> |Fetch Avoid|35±1.9|36±2.8|37±1.5|13.7±14.8|35.7±5.5|34.9±6.7|12.1±8.8|
> |Fetch Align|30.1±6.4|21±15.9|20±20|1.8±3.3|0±0|0±0|0±0|
> |Carlo|2.6±1.1|3.2±1.8|2.3±0.3|0±0|0.3±0.5|0±0|0±0|
> |CheetahFlip|7.25±1.3|3.5±2.1|3.3±3.3|0±0|2.5±5|2.4±1.6|0±0|
>
> All our ablations outperform the baselines. For easy tasks like GridCircular Hard and FlatWorld Stabilization, this hyperparameter doesn't have a significant impact. Some challenging tasks such as Fetch Align and CheetahFlip perform better with larger $\beta$, meaning the shaped reward from the discriminator guides the learning more effectively.
>
> **Q2.2 Additionally, since the reward depends on the length of the longest acyclic path, could the authors provide more details on how this is computed and how it impacts learning, especially in cases where the path is long, which may lead to very small or vanishing reward signals?**
>
> We compute the longest acyclic path by performing depth-first search to enumerate all simple paths from a given automaton state to any accepting state. If a state is encountered that is already on the current path, we do not extend the path further through that state. Since LDBAs are typically small and structurally constrained in our benchmarks, this approach remains computationally tractable in all tasks we considered. Normalizing the progress-based reward in Eq. 8 by the length of the longest path is beneficial for stabilizing training as it ensures that all progress signals are kept within a consistent [0,1] range, which helps maintain reward sensitivity throughout training. Empirically, we did not observe vanishing gradient issues—likely due to the sparse nature of LDBAs.
>
> **Q3. The proposed method assumes access to expert demonstrations. Could the authors clarify the underlying assumptions about these demonstrations? Specifically, do the demonstrations contain trajectories that visit each accepting state, and do they provide sufficient coverage of the state space? For instance, in the FlatWorld Cycle environment, if the demonstrations only reach the yellow region, how would the method support learning policies that reach the green region? Similarly, if all demonstrations originate from the left side of the environment, would this limit the agent’s ability to explore or learn effectively when initialized from the right side?**
>
> * TiLoIL assumes that the demonstrations cover key transitions of an LDBA, especially by visiting accepting states at least once or twice  (Line 184), thereby enabling it to identify and learn from successful automaton progressions. We provide a detailed discussion and examples of this assumption in our response to Reviewer Tszg (Q2).
>
> * However, TiLoIL does *not* require the demonstrations to provide sufficient coverage of the state space because it is able to leverage the agent's own behaviors—those that have progressed beyond certain LDBA states—as positive trajectories to train discriminators that guide learning for those trajectories that fail to meet such LDBA states, thereby reducing reliance on large demonstration datasets (Fig. 4 further supports this, as TiLoIL_noStage, which lacks multi-stage discriminators and resembles GAIL, performs worse).  In FlatWorld Cycle, if we use the demonstrations that originate from the left side of the environment, and train the agent starting from the right, it still learns well (converging to 3 returns after 200K steps).
>
> * Importantly, TiLoIL does *not* require user-demonstrations to be of high quality. In practice, user-provided demonstrations may be suboptimal due to the cost and complexity of constructing trajectories that robustly satisfy such objectives. In our setup, demonstrations are generated by chaining locally trained controllers for each stage of the LTL task (line 341). These policies are not globally optimal: when executed repeatedly, they often fail to produce trajectories that visit LDBA-accepting states infinitely often. For example, in CheetahFlip, the chained controllers—and thus the resulting demonstrations—achieve only one or two rotations, since additional flips require maintaining momentum after reaching an accepting state, which the learned local policies cannot sustain. Similarly, in Carlo, demonstrations rarely reach accepting states more than twice. Despite this suboptimality, TiLoIL learns policies that outperform the demonstrations, empirically visiting accepting states more frequently in both CheetahFlip and Carlo.
>
> **Q4. In the experiments, how many episodes are averaged per evaluation (in addition to the 10 random seeds) for each environment? Also, aside from Cheetah Flip, are the initial positions of the agents and the placement of regions or objects randomly generated during evaluation?**
>
> We evaluated each policy using 10 episodes per random seed, averaged across 10 seeds. The initial positions are random in Doggo, Fetch, and Carlo environments; fixed in FlatWorld environments.

---

> > ### Comment · Reviewer_uyFt · 2025-08-01
> >
> > I appreciate the authors’ detailed responses, and most of my concerns have been addressed. However, my primary concerns regarding scalability and the assumptions underlying the demonstrations remain.
> >
> > Regarding the scalability, the authors note that “Particularly, the number of accepting states in an LDBA is small—often just one in our benchmarks—and does not scale with specification complexity. In practice, all benchmark LTL specifications in our work (and related studies) produce LDBAs with ≤ 10 states, resulting in only a small number of discriminators and computationally feasible training” While this observation may hold for the evaluated benchmarks, it does not necessarily generalize to more complex specifications, which can yield LDBAs with large number of states and multiple accepting conditions. The current evaluation appears restricted to relatively simple specifications.
> >
> > Regarding the assumptions on demonstrations, the method requires that “TiLoIL assumes that the demonstrations cover key transitions of an LDBA, especially by visiting accepting states at least once or twice,” and that “demonstrations must sufficiently cover the behavioral variations necessary to reach the accepting conditions under different modes” (response to Reviewer Tszg (Q2)). These assumptions are strong and may limit the method’s applicability. As I mentioned above, for complex LTL specifications, collecting demonstrations that adequately cover key transitions of the corresponding LDBA can be challenging in practice. Providing empirical results on such specifications would substantially strengthen the paper and better demonstrate the effectiveness of the proposed approach.
> >
> > In addition, regarding the reward function $R((s, b))$, the authors state that “although $R((s, b))$ does not directly distinguish between actions, the transition dynamics do: different actions from the same state lead to different future product states, resulting in different expected cumulative rewards.” However, this statement is unclear. Since the MDP is defined with a transition probability function $P$ (line 102), different actions from the same state can lead to the same future product states. Is the transition function $P$ assumed to be deterministic?

---

> > > ### Author Response · Authors · 2025-08-04
> > > **Addressing the scalability issue**
> > >
> > > **Regarding the scalability, ..., it does not necessarily generalize to more complex specifications, which can yield LDBAs with large number of states and multiple accepting conditions.**
> > >
> > > We agree that more complex LTL specifications can increase the number of discriminators, making training more computationally intensive and potentially unstable.
> > >
> > > *We have conducted additional experiments to address this scalability concern.* Recall that our reward formulation in the paper is $R((s,b)) = \frac{\text{SIdx}(b) + \beta \cdot \tanh(f_b(s))}{\mathcal{N}(b)}$, where $\text{SIdx}(b)$ serves as an approximation of the stage index of the product MDP state $(s, b)$ (Line 303), $\beta$ is a hyperparameter (see response Q2.1), and $f_b(s)$ is a discriminator associated with LDBA state $b$. To mitigate the scalability issue of training a separate discriminator per LDBA state, we experiment with a *single global discriminator* $f(b, s)$, which is conditioned on the current LDBA state $b$, and reformulate the reward as $R((s,b)) = \frac{\text{SIdx}(b) + \beta \cdot \tanh(f(b, s))}{\mathcal{N}(b)}$. In implementing $f(b, s)$, we concatenate a one-hot encoding $z_b \in \mathbb{R}^K$ of the LDBA state $b$ (where $K$ is the number of LDBA states) with the state vector $s$ as input to the discriminator network. As in our original implementation, we train $f(b, s)$ to predict whether an MDP state $s \in S$ originates from a positive trajectory whose maximal stage exceeds the LDBA state $b$, as opposed to a negative trajectory that fails to progress beyond $b$ or gets trapped in a sink state.
> > >
> > > Surprisingly, the shared global discriminator *outperforms* the original per-state discriminators, as measured by eventual discounted returns over 5 random seeds.
> > > |Env|TiLoIL(multiple discriminators)|TiLoIL (shared discriminator)|
> > > |-|-|-|
> > > |Doggo Avoid|296±74|360±20|
> > > |Doggo Navigate|246±61|286±91|
> > > |Fetch Avoid|36±2.8|37±0.7|
> > > |Fetch Align|21±15.9|26±22|
> > > |Carlo|3.2±1.8|3.9±0.5||
> > > |CheetahFlip|3.5±2.1|3.5±2.5|
> > > |FlatWorld Cycle|2.3±0.8|2.8±0.1|
> > >
> > > We hypothesize that this is due to more efficient data sharing across LDBA states, which stabilizes training and prevents overfitting to individual LDBA states that may receive fewer positive samples. This result suggests that structured conditioning of a shared discriminator is a promising future direction to improve scalability without compromising performance (for example, one could future explore introducing FiLM layers that modulate hidden activations in the shared discriminator based on the LDBA state). We will include this result in the revised paper.

---

> > > > ### Author Response · Authors · 2025-08-04
> > > > **Addressing the assumptions on demonstrations**
> > > >
> > > > **Regarding the assumptions on demonstrations, ... collecting demonstrations that adequately cover key transitions of the corresponding LDBA can be challenging in practice.**
> > > >
> > > > Our primary contribution is to show that finite-length, non-optimal demonstrations—collected by suboptimal policies that reach LDBA accepting states only once or twice—can be leveraged to learn policies that satisfy LTL objectives over infinite horizons. The key lies in generalizing from such limited behaviors to policies that reach accepting states infinitely often. If demonstrations fail to cover certain transitions to LDBA accepting states, it becomes the responsibility of the underlying RL algorithm (SAC in TiLoIL) to discover trajectories that reach those states through its own exploration. TiLoIL is able to leverage the agent's own behaviors—those that have progressed beyond certain LDBA states—as positive trajectories to train discriminators that guide learning for those trajectories that fail to meet such LDBA states (Line 14, Algorithm 1), thereby reducing reliance on large demonstration datasets. However, if the RL algorithm fails to explore LDBA accepting states, TiLoIL is likely to fail. We will strengthen the assumption stated in line 184 and acknowledge this limitation in the revision.
> > > >
> > > > A promising future direction is to combine TiLoIL’s exploitation strategy—which leverages demonstrations—with advanced exploration techniques from recent RL-for-LTL algorithms, aiming to overcome the reliance on demonstrations as a bottleneck. To investigate this idea, we implemented TiLoIL$^{+}$, which integrates TiLoIL with the state-of-the-art exploration technique in DRL2. Notably, TiLoIL$^{+}$ does not use demonstrations. Instead, it simply extends TiLoIL's reward function $R((s,b))$ to $R_{c}((s,b), (s',b')) = R((s',b')) + R_\text{intr}(b, b')$, where $(s',b')$ is the next product MDP state and $R_{\text{intr}}(b, b')$ is an intrinsic reward based on potential-based shaping for each LDBA transition, as in DRL2. The results are measured by eventual discounted returns over 5 random seeds after 1M training steps:
> > > > |Env|TiLoIL|DRL2|TiLoIL$^{+}$|
> > > > |-|-|-|-|
> > > > |Doggo Avoid|296±74|79.9±67.1|220±75|
> > > > |Doggo Navigate|246±61|45.9±64.7|133±93|
> > > > |Fetch Avoid|36±2.8|35.7±5.5|37.9±0.3|
> > > > |Fetch Align|21±15.9|15±18*|27±16|
> > > > |Carlo|3.2±1.8|0.3±0.5|1.9±1.6|
> > > > |CheetahFlip|3.5±2.1|2.5±5|20±1.75|
> > > >
> > > > *DRL2 fails to achieve non-zero reward by 1M steps; we report its best performance within this interval.
> > > >
> > > > TiLoIL$^{+}$ significantly outperforms DRL2. While exploration alone (in DRL2) may only yield agents with low to medium success rates, combining this with TiLoIL's exploitation allows the agent to learn more effectively from its own experience—reinforcing behaviors that lead to progress on the LTL objective. Remarkably, TiLoIL$^{+}$ performs close to TiLoIL, despite lacking demonstrations. This result shows that integrating TiLoIL's exploitation with directed exploration can help reach LDBA transitions that are otherwise difficult to discover through naïve exploration alone. Although this result lies beyond the scope of our current paper, we believe TiLoIL opens an exciting direction—systematically combining exploitation and exploration to enable efficient learning of LTL-constrained policies without demonstrations as a limiting factor.
> > > >
> > > > [DRL2] Directed Exploration in Reinforcement Learning from Linear Temporal Logic. Transactions on Machine Learning Research. 2025.
> > > >
> > > > **Since the MDP is defined with a transition probability function, different actions from the same state can lead to the same future product states. Is the transition function assumed to be deterministic?**
> > > >
> > > > We agree the phrasing can be clarified. By "different actions lead to different future product states," we did not mean all actions must lead to distinct next states. Rather, we mean that actions can differ in their distribution over future product states, which affects the expected cumulative reward. The transition function is not assumed to be deterministic.
> > > >
> > > > We sincerely thank the reviewer for their thoughtful feedback and continued engagement in improving our work!

---

> > > > > ### Comment · Reviewer_uyFt · 2025-08-05
> > > > >
> > > > > I appreciate the authors' clarifications and additional experiments. However, I remain concerned about the scalability of the proposed method with respect to the complexity of LTL specifications, particularly those that induce a large number of LDBA states and multiple accepting states. While the provided results are interesting, they are still limited to relatively simple specifications, and it remains unclear how the method would perform on more complex LTL tasks.

---

> > > > > > ### Author Response · Authors · 2025-08-07
> > > > > > **Thank you for your feedback!**
> > > > > >
> > > > > > **How TiLoIL would perform on more complex LTL tasks?**
> > > > > >
> > > > > > (1) In the appendix, we evaluated TiLoIL against a disjunctive LTL formula $(F(y \land XFg) \vee F(r \land XFg)) \land G \neg b$, which features multiple accepting conditions, in the Flatworld environment. Please refer to Fig. 9, Line 792. This task requires reaching (yellow → green) or (red → green) while avoiding blue. Demonstrations include trajectories for both paths. We evaluated performance using eventual discounted returns over 10 seeds.
> > > > > >
> > > > > > | Method                | LCER       | DRL2      | GAIL      | TiLoIL     |
> > > > > > | --------------------- | ---------- | --------- | --------- | ---------- |
> > > > > > | Flatworld Disjunction | 42.9 ± 0.1 | 0.0 ± 0.0 | 5.0 ± 9.9 | 43.5 ± 0.3 |
> > > > > >
> > > > > > While LCER achieves similar eventual discounted returns as TiLoIL, it fails to adapt its path based on the initial state, always choosing yellow → green. In contrast, TiLoIL dynamically selects the optimal path, e.g., choosing red → green if the agent starts closer to red, and vice versa.
> > > > > >
> > > > > > (2) We deployed the FlatWorld agent in the 16Rooms environment borrowed from [51], where task $\Psi$ requires the agent to traverse a circular path indefinitely via one of two possible routes. The rooms are arranged in a $4 \times 4$ grid, and are indexed as $\texttt{room}\_{i,j}$ with $i$ denoting the row index (from left to right) and $j$ the column index (from bottom to top).
> > > > > >
> > > > > > The LTL formula:
> > > > > > $\Psi = \Big(
> > > > > > \mathbf{GF}(\texttt{room}\_{1,2} \wedge
> > > > > > \mathbf{XF}(\texttt{room}\_{1,4} \wedge
> > > > > > \mathbf{XF}(\texttt{room}\_{3,4} \wedge
> > > > > > \mathbf{XF}(\texttt{room}\_{3,2}))))
> > > > > > \vee \mathbf{GF}(\texttt{room}\_{3,1} \wedge
> > > > > > \mathbf{XF}(\texttt{room}\_{3,3} \wedge
> > > > > > \mathbf{XF}(\texttt{room}\_{3,4} \wedge
> > > > > > \mathbf{XF}(\texttt{room}\_{4,4} \wedge
> > > > > > \mathbf{XF}(\texttt{room}\_{4,3})))))
> > > > > > \Big)$
> > > > > > $\wedge \mathbf{G}(\neg \texttt{Wall})$
> > > > > > specifies that the agent must repeatedly visit either one of two possible loops:
> > > > > > * The large loop goes through rooms $(1,2) \rightarrow (1,4) \rightarrow (3,4) \rightarrow (3,2)$.
> > > > > > * The small loop goes through rooms $(3,3) \rightarrow (3,4) \rightarrow (4,4) \rightarrow (4,3)$.
> > > > > > * The agent must avoid collisions with wall segments between rooms.
> > > > > >
> > > > > > The optimal policy needs to identify and traverse the smaller loop repeatedly while avoiding all walls. The disjunction of two nested $\mathbf{GF}$-$\mathbf{XF}$ chains encodes alternative loops through specific rooms, introducing deep temporal nesting and alternation. This leads to a state explosion in the LDBA, as it must track partial progress along multiple paths while enforcing ordering and recurrence constraints. The combination of eventuality and next operators within a disjunction further compounds complexity by requiring concurrent tracking modes.
> > > > > >
> > > > > > We evaluated our method and the baselines in this environment and found that TiLoIL is the only method that reliably learns policies satisfying $\Psi$, consistently traversing the small loop (the demonstrations include trajectories for each loop). Other baselines fail to make progress due to the intricate LDBA structure.
> > > > > >
> > > > > > (3) In our response to Q2 for Reviewer Tszg, we explored both a Cookie-like task and a Symbol-like task based on the Fetch Pick&Place environment, with complex LTL objectives. Notably, the Symbol-like task induces an LDBA with two accepting conditions.
> > > > > >
> > > > > > We will include all these environments in the revised version to address your concern. Thank you for the suggestion!
> > > > > >
> > > > > > **LDBA Complexity Does Not Always Correlate with Task Difficulty**
> > > > > >
> > > > > > We would also like to clarify that higher LDBA complexity does not necessarily imply a more difficult task. As acknowledged in DRL2 [6], exploration becomes significantly more challenging in environments with long horizons and rare LDBA transitions, due to the difficulty in triggering those transitions. In the Cheetah environment, for example, the agent can be guided by the LTL formula GF(a ∧ XF(b ∧ XF(c ∧ XF d))), where each atomic proposition corresponds to a specific pose of the robot—default posture (a), standing on front legs (b), upside down (c), and standing on back legs (d). This objective induces a complex LDBA with multiple transitions. However, our main evaluation uses the simpler formula GF(b ∧ XF d), which results in fewer LDBA transitions. Nevertheless, as shown in the table below, the more complex objective actually leads to better performance for DRL2, suggesting that the simpler formula is harder due to sparse and hard-to-trigger transitions. This highlights that task difficulty depends not only on the LDBA's size or structure but also on how transition triggering dynamics interact with the environment, as reflected in our benchmarks.
> > > > > >
> > > > > > | Env                                        | TiLoIL    | DRL2     |
> > > > > > | ------------------------------------------ | --------- | --------- |
> > > > > > | CheetahFlip `GF(b ∧ XF d)`                 | 3.5 ± 2.1 | 2.5 ± 5   |
> > > > > > | CheetahFlip `GF(a ∧ XF(b ∧ XF(c ∧ XF d)))` | 3.7 ± 0.8 | 3.3 ± 6.6 |

---

> > > > > > > ### Author Response · Authors · 2025-08-08
> > > > > > > **Planned Revisions**
> > > > > > >
> > > > > > > Thank you once again for your helpful feedback. We will address your concerns in the final version through the following revisions:
> > > > > > > * To address your concern about scalability, we will include results on Flatworld Disjunction, 16Rooms, the Cookie-like, and Symbol-like tasks (see our earlier reply for details) to demonstrate TiLoIL's performance on complex LTL specifications. Additionally, we will use the CheetahFlip example from our earlier reply to illustrate the relationship between LDBA complexity and task difficulty. We will also report preliminary results (from our earlier response) suggesting that structured conditioning of a shared discriminator is a promising future direction to further resolve the scalability issue.
> > > > > > > * To address your concern about the assumptions on demonstrations, we will clarify that if demonstrations fail to cover certain transitions to LDBA accepting states, it becomes the responsibility of the underlying RL algorithm to discover trajectories that reach those states through its own exploration. While this is a limitation, TiLoIL is still able to leverage the agent's own behaviors—those that have progressed beyond certain LDBA states—as positive trajectories to train discriminators that guide learning for those trajectories that fail to meet such LDBA states, thereby reducing reliance on large demonstration datasets. To further validate this, we conducted an additional experiment in the Carlo environment. This environment involves a self-driving agent trained to follow a circular track counterclockwise while visiting two blue regions repeatedly and avoiding crashes. We trained TiLoIL using only failed or incomplete demonstrations, including those that either entered unsafe regions or failed to complete a full loop by missing one blue region. Results (averaged over 5 seeds after 400K training steps) are shown below. While there is a performance drop—as expected—TiLoIL remains effective, significantly outperforming the RL for LTL baselines and GAIL, as reported in the main paper. We will include this discussion and the results in the revised version to clarify the method's robustness under such suboptimal demonstrations.
> > > > > > > |Environment|	TiLoIL (Original Demos)|	TiLoIL (Unsafe or Incomplete Demos Only)|
> > > > > > > |-|-|-|
> > > > > > > |Carlo|	3.2 ± 1.8|	2.1 ± 0.5|
> > > > > > > * We will clarify that our reward function depends only on states because specification satisfaction—the objective being optimized—is determined solely by the product state, not the specific action taken.
> > > > > > > * We will include the ablation study on $\beta$ to illustrate how TiLoIL uses it to control the influence of shaped rewards from multi-stage discriminators, and demonstrate that the method maintains strong performance across different values of $\beta$.
> > > > > > > * We will elaborate on how we compute the length of the longest acyclic path in the LDBA and explain its role in stabilizing learning.
> > > > > > > * We will clarify our environment setup, including the number of evaluation episodes and the degree of randomness in initial states, to ensure reproducibility of our results.
> > > > > > >
> > > > > > > We sincerely appreciate your detailed feedback, which has been invaluable in improving both the clarity and technical depth of our work!

---

> > ### Comment · Reviewer_uyFt · 2025-08-09
> >
> > Regarding Q4, just to confirm, are both the agents’ initial positions and the placement of regions or objects randomly generated during training and evaluation?

---

### Official Review · Reviewer_ifPj · 2025-07-03

**Clarity:** 2
**Significance:** 3
**Originality:** 3
**Rating:** 4
**Confidence:** 4

**Summary:**

This work presents TiLoIL, an imitation learning method for achieving objectives specified using Linear Temporal Logic (LTL). The key innovations are two forms of decomposition: first, segmenting expert trajectories to learn how to reach accepting states in a corresponding LDBA, and second, breaking down tasks into stages to facilitate learning.

**Questions:**

Please see the points under weaknesses for more detailed discussion. The main questions are:

- Are there any implicit objectives in the demonstrations beyond the LTL specifications?
- How should TiLoIL be categorized in relation to existing work (e.g., online vs. offline, imitation learning vs. reinforcement learning, or a hybrid)?
- How are the product MDPs constructed, particularly in continuous domains?
- What kinds of tasks is TiLoIL likely to succeed or struggle with, and why?

I am willing to adjust my score depending on the responses to the above.

**Ethical Concerns:**

["NO or VERY MINOR ethics concerns only"]

**Final Justification:**

The authors responses helped to clarify the problem setup and situate the work relative to existing methods on learning with LTL objectives/constraints. I do think the work contributes towards an important area. On the downside, the work has limited significance and technical novelty (a subjective opinion) hence not a stronger accept recommendation.

**Limitations:**

Yes

**Quality:**

2

**Strengths And Weaknesses:**

### Strengths:

- Enabling agents to follow temporally extended symbolic objectives is an important problem.
- Clearly identifies the challenge of using finite trajectories for general LTL rather than resorting to LTLf (LTL over finite traces).
- The proposed methodology appears sound, and experimental results demonstrate effectiveness on selected tasks.

### Weaknesses:

- **Clarity of formulation and demonstrations**: The formulation could be clearer about the nature of the expert demonstrations and how they relate to the LTL objectives. Why don't *all* expert demonstration satisfy the given LTL? The setup is presented as multi-objective, but it seems the primary goal is to find a policy that satisfies LTL constraints, unless there are additional implicit objectives encoded in the demonstrations. Clarification would help.

- **Positioning within the literature**: The related work could be better situated. The second paragraph of Appendix B should be included in the main paper and expanded to clarify the similarities and differences with existing approaches. For example, from my understanding, TiLoIL requires environment interaction, whereas [20] operates in a purely offline setting. [20] also uses trajectory segmentation and does not require expert demonstrations, something noted as future work for TiLoIL. Similar comparisons should be made against the other prior work in a more systematic fashion, e.g., by categorizing the methods.

- **Use of LDBAs in continuous tasks**: It is unclear how LDBAs and product MDPs are constructed and used for tasks with continuous state spaces. Was the state space discretized? Given that LDBAs are worst-case exponential in the size of the LTL formula, this may limit scalability to more complex objectives.

- **Experimental setup and insights**: While TiLoIL performs well on many tasks, the task selection could be more clearly motivated. For example, Minecraft and Pacman appear in [63], while [6] includes two Cheetah variants. Clarifying why these tasks were chosen would help us understand the results. Additionally, it would be valuable to discuss the types of tasks where TiLoIL is likely to succeed or fail, to guide future research. On the Cheetah Flip task, the multi-stage approach appears to have hindered learning. The explanation offered is a good start, but a deeper analysis of failure cases would improve understanding and potential resolutions.

---

> ### Author Rebuttal · Authors · 2025-07-29
>
> We appreciate your insightful feedback and constructive comments!
>
> **Q1. Are there any implicit objectives in the demonstrations beyond the LTL specifications?**
>
> **W1. Why don't all expert demonstration satisfy the given LTL? The setup is presented as multi-objective, but it seems the primary goal is to find a policy that satisfies LTL constraints, unless there are additional implicit objectives encoded in the demonstrations.**
>
> Our approach (TiLoIL) is developed for Imitation Learning *with Temporal Logic Constraints*. For imitation learning, one can simply optimize $\pi^\ast = \arg \max_{\pi_\phi \in \Pi} J_\pi(\phi) $ where $J_\pi(\phi)$ is the GAIL objective minimizing the Jensen-Shannon) divergence between the learning policy and expert policy (Line 159). However, in practice, user-provided demonstrations may be suboptimal due to the cost and complexity of constructing them. Learning solely from finite-length demonstrations does not guarantee deriving a policy that generalizes to infinite horizon LTL tasks. As a concrete example, in our setup, demonstrations are generated by chaining locally trained controllers for each stage of the LTL task (line 341). These policies are not globally optimal: when executed repeatedly, they often fail to produce trajectories that visit LDBA-accepting states infinitely often. In CheetahFlip, the chained controllers—and thus the resulting demonstrations—achieve only one or two rotations, since additional flips require maintaining momentum after reaching an accepting state, which the learned local policies cannot sustain. Similarly, in Carlo, demonstrations rarely reach accepting states more than twice. To address this issue, we add LTL constraints forming a multi-objective learning paradigm: $\pi^\ast = \arg \max_{\pi_\phi \in \Pi} \left( P(\pi_\phi \models \varphi), J_\pi(\phi) \right)$. The agent not only imitates demonstrations but also maximizes the probability of LTL satisfaction. Under this formalism, despite demonstration suboptimality, TiLoIL learns policies that outperform the demonstrations, empirically visiting accepting states more frequently in both CheetahFlip and Carlo.
>
> Although demonstrations do not encode implicit objectives, the multi-objective formulation—combining demonstrations with LTL specifications—provides a compelling advantage: it enables policies to be learned significantly faster than with pure RL, while requiring far fewer demonstrations than classical imitation learning.
>
> **Q2. How should TiLoIL be categorized in relation to existing work (e.g., online vs. offline, imitation learning vs. reinforcement learning, or a hybrid)?**
>
> TiLoIL is a hybrid online imitation learning method. It extends the adversarial imitation learning framework of GAIL to tasks specified by LTL objectives. Like GAIL, it performs online interaction with the environment while training a discriminator on expert demonstrations to provide a shaped reward signal. The demonstrations are used to guide exploration in the early stages, but the policy is improved via reinforcement learning beyond the demonstration quality. Thus, TiLoIL is neither a purely offline method nor standard behavioral cloning—it is an online, adversarial imitation learning approach specialized for structured, infinite-horizon LTL tasks. The primary contribution is to show that *finite-length* demonstrations can be used to overcome the exploration bottleneck inherent in learning LTL objectives over *infinite* horizons.
>
> **Q3. How are the product MDPs constructed, particularly in continuous domains?**
>
> **W3. Use of LDBAs in continuous tasks: Was the state space discretized? Given that LDBAs are worst-case exponential in the size of the LTL formula, this may limit scalability to more complex objectives.**
>
> As is standard in RL for LTL tasks, we do not explicitly construct the full product MDP, which would be infeasible in continuous domains due to the infinite state space. Instead, we simulate product MDP transitions on-the-fly: at each environment step, we maintain a product state $(s, b) \in \mathcal{S} \times \mathcal{B}$, where $s$ is the current (continuous) environment state and $b$ is the current LDBA state. Upon taking an action, the environment transitions to a new state $s'$ ($s = s'$ if the action is an epsilon transition), and the automaton transitions to $b'$ according to the LDBA transition relation and the atomic propositions satisfied by $s'$ (Def. 2.3). This approach avoids constructing the full product MDP while preserving correctness, enabling us to learn policies over $\mathcal{S} \times \mathcal{B}$ that satisfy the LTL objective in a *model-free* manner. Crucially, we do not discretize the state space, as such discretization is computationally prohibitive and does not scale to high-dimensional environments.
>
> **Q4. What kinds of tasks is TiLoIL likely to succeed or struggle with, and why?**
>
> **W4. Experimental setup and insights: While TiLoIL performs well on many tasks, the task selection could be more clearly motivated.**
>
> TiLoIL has been tested on all the benchmark environments used in prior work LCER [63] and DRL2 [6]. We exclude Minecraft and Pacman, as these are simple, low-dimensional discrete environments where exploration is not challenging. In fact, TiLoIL can solve these tasks even without demonstrations—random exploration is sufficient to encounter accepting states, enabling TiLoIL to learn a policy with performance comparable to the LCER baseline [63]. We do include the CheetahFlip environment from [6], while omitting the other Cheetah variant, which is structurally similar to DoggoNavigate in both dynamics and LTL structure. In the experiment section, our focus is on evaluating TiLoIL in tasks that involve nontrivial exploration challenges and diverse LTL objectives.
>
> Overall, TiLoIL performs better when demonstrations are of higher quality. As noted in our response to Q1, TiLoIL's relatively weaker performance on CheetahFlip—compared to its ablated variant, though still outperforming all baselines—is primarily due to suboptimal demonstrations. These demonstrations can only achieve one or two rotations and fail to provide sufficient supervision for maintaining the momentum needed to consistently flip while standing on the back leg. Despite this limitation, TiLoIL surpasses the demonstrations, suggesting that its performance is bottlenecked by the quality of the demonstrations rather than its learning capability. In contrast, the ablated variant TiLoIL_noStage focuses on learning to repeatedly stand on the front legs, a posture from which maintaining or regaining forward rotational momentum is easier due to the task's dynamics.
>
> TiLoIL is especially effective in tasks where reaching accepting states requires structured, multi-stage behavior, and where random exploration is insufficient. In such settings, even suboptimal demonstrations that partially reach accepting conditions can significantly help exploration. Conversely, TiLoIL may struggle in tasks where demonstrations fail to sufficiently cover the key transitions in the automaton (e.g., CheetahFlip), or when the LTL formula requires environmental events that demonstrations fail to expose. Overall, TiLoIL’s performance is influenced by both the quality of demonstrations and their coverage of the automaton's structure.
>
> **W2. Positioning within the literature**
>
> We will revise the paper to move the second paragraph of Appendix B into the main text and expand it to more clearly position TiLoIL in relation to existing approaches. For example, Temporal Logic Imitation integrates LTL-based task specifications with low-level control but assumes a pre-trained continuous controller, limiting its ability to improve low-level behavior. In contrast, TiLoIL jointly learns both high-level logic alignment and low-level control from scratch in an end-to-end manner. Similarly, Diffusion Meets Options, LTLDoG, and CTG are fully offline methods trained on large datasets (e.g., 6,686 trajectories in PushT), whereas TiLoIL operates in an interactive setting and learns from only a handful of expert trajectories (e.g., 5 in Carlo). This makes direct empirical comparisons challenging, but highlights TiLoIL’s efficiency in low-data regimes with structured supervision. We will also expand our discussion to more systematically categorize prior work by (1) their strategies to leverage LDBA structures to address the exploration challenge, (2) whether they leverage automaton structures for hierarchical control, and (3) whether they operate in an online or offline setting.

---

> > ### Comment · Reviewer_ifPj · 2025-08-05
> > **Thanks for the response.**
> >
> > Thank you to the authors for the response. I better understand the contribution and its positioning, as well as the choices for experimental domains.

---

> > > ### Author Response · Authors · 2025-08-05
> > > **Thank you for your follow-up!**
> > >
> > > Thank you again for your engagement during the discussion. We are glad the response helped clarify the contribution and experimental design. We will revise the final version to address your concerns as follows:
> > >
> > > * We will expand the discussion around Line 133 to clearly state the assumptions TiLoIL makes about demonstrations, emphasizing that they may be suboptimal and do not encode implicit objectives beyond the LTL specification. We will also clarify the role of the multi-objective formulation in overcoming the limitations of such demonstrations. This clarification will help elucidate the types of tasks where TiLoIL is likely to succeed or struggle.
> > >
> > > * We will revise and relocate content from Appendix B to the main paper, expanding our comparison with existing approaches (e.g., Temporal Logic Imitation, Diffusion Meets Options, CTG, LTLDoG). This will clarify TiLoIL's hybrid nature—as an online adversarial imitation learning method—and highlight its distinct contributions in the context of structured LTL supervision and limited demonstrations.
> > >
> > > * We will clarify that we do not construct the full product MDP in continuous domains, but simulate product transitions on-the-fly, avoiding state space discretization while preserving correctness.
> > >
> > > * We will explain our task selection rationale more clearly, emphasizing TiLoIL’s strength in settings with challenging exploration and structured LTL specifications.
> > >
> > > Your insights have been invaluable in helping us strengthen the paper. If there’s anything further we can clarify or improve to support a more positive assessment, we would be truly grateful for your feedback.

---

> > > > ### Comment · Reviewer_ifPj · 2025-08-08
> > > > **thanks for the summary**
> > > >
> > > > Thanks for summarizing the revisions. I have no further comments as the response was clear to me.
> > > >
> > > > The clarification regarding the "multi-objective" formulation was especially helpful. I still think that using the term "multi-objective" may be misleading; multi-objective optimization often deals with conflicting objectives, which is not the case here. There is really just one objective but with two terms; one corresponding to the imitation loss and the other corresponding to the LTL adherence. Perhaps the authors should just avoid using the term but this is a minor quibble.

---

> > > > > ### Author Response · Authors · 2025-08-08
> > > > > **Thank you for your suggestion!**
> > > > >
> > > > > Thank you for your thoughtful feedback and for finding our clarification helpful. Following your suggestion, we will revise the paper to avoid the term "multi-objective". Instead, we will clarify that the agent is trained using a joint objective that simultaneously optimizes imitation and LTL satisfaction, which we believe more accurately reflects our formulation. We appreciate your close reading and helpful input!

---

### Official Review · Reviewer_r1Rp · 2025-07-03

**Clarity:** 3
**Significance:** 3
**Originality:** 3
**Rating:** 5
**Confidence:** 3

**Summary:**

The paper deals with the problem of imitation learning where the learning agent has access to both a set of positive expert demonstrations, and in addition a task specification in LTL which the expert demonstrations satisfy. Since LTL is evaluated over infinite trajectories, RL approaches design proxy rewards based on accepting states (an LTL formula can be translated to a LDBA state machine, where an infinite trajectory satisfies the LTL if it visits an accepting state infinitely often). The inherent difficulty in this approach is that the rewards are sparse, making exploration difficult. Expert demonstrations can theoretically help, but are by nature finite.
To address these difficulties the authors splits trajectories into "segments", sub-trajectories between accepting states (these are effectively treated as independent demonstrations), and within those segments they leverage the structure of the LTL formula (as evident in the LDBA) to further recognize "stages" of satisfaction of the formula, allowing them to densify the rewards. Their approach falls within the framework of GAIL, alternately training a discriminator to distinguish between states from (suitably defined) positive and negative sub-trajectories, and using the learned discriminator(s) to define dense rewards for policy improvement. They optimize for both satisfaction of the formula (reaching an accepting state) and IL goal of making the policy trajectories indistinguishable from the expert demonstrations. They evaluate their learned policies on a number of standard tasks in high-dimensional spaces (some of them circular, to specifically evaluate the merits of their segmentation method), and test two aspects of their work - comparing it to SOTA policy optimization methods that make use of LTL objectives (but not expert demonstrations) to see if it improves exploration, and to IL methods that do not explicitly leverage LTL specifications.

**Questions:**

- I understand that non-optimal demonstrations are out of scope, but given that in the real world they exist, what is the effect of adding such demonstrations in practice? Will the results degrade gracefully, or can one sub-optimal demonstration, no matter how close to satisfaction, collapse the entire optimization?

- If the epsilon-transitions have any meaning in your framework, I'd love to see them incorporated into an example (The flatworld perhaps)

- Can this method be adapted to a setting where the observations of the environment are noisy/partially observed? What if I'm only 90% certain I hit 'blue'? I realize it is out of scope, but addressing this in the discussion/limitations would add broader context.

**Ethical Concerns:**

["NO or VERY MINOR ethics concerns only"]

**Final Justification:**

I stay with the 5 score.

The authors thoroughly answered my questions, and this along with their discussions with other reviewers helped me get a better picture regarding the effect of different sub-optimal demonstrations. The other issues were related to clarity of their contributions, and I accept that they will make the appropriate changes.

**Limitations:**

The authors address the main limitation - that the method assumes optimal demonstrations.

**Paper Formatting Concerns:**

No major issues.

**Quality:**

3

**Strengths And Weaknesses:**

Strengths:

- The paper is novel in that it combines the LDBA-MDP product formulation with expert demonstrations, combining global task specification with more local guidance within the segments from the demonstrations.

- The evaluation is presented clearly, the ablation results elucidate the importance of their segmentation technique in cyclical tasks. The results seem good compared to existing SOTA methods.

- The paper is well-written, and related work and context is thoroughly explored.

Weaknesses:

- The contributions of the paper are not clearly outlined. As a non-expert in the field, it took me a while to (hopefully) disentangle the issues of infinite horizon and reward sparsity.

- While epsilon-transitions in the LDBA are formally defined, I struggled to understand from the paper alone what their meaning is in the product MDP. The running example makes no use of them, their relation to deterministic vs non-deterministic states is not mentioned and overall I'm not sure why they were introduced (other than for completeness). Its not a major issue, but for a non-expert it was a sticking point.

---

> ### Author Rebuttal · Authors · 2025-07-29
>
> We appreciate your insightful feedback and constructive comments!
>
> **Q1. What is the effect of adding non-optimal demonstrations in practice? Will the results degrade gracefully, or can one sub-optimal demonstration, no matter how close to satisfaction, collapse the entire optimization?**
>
> We clarify that the assumption of expert demonstrations applies only to our theoretical analysis (e.g., Theorem 3.1). In practice (line 341), our demonstrations are generated by chaining locally trained controllers for each stage of the LTL task. The policies learned in this way are not optimal: when executed repeatedly, they fail to produce trajectories that visit LDBA-accepting states infinitely often. For example, in CheetahFlip, the chained controllers—and thus the collected demonstrations—achieve only one or two rotations, since additional flips require maintaining momentum after reaching an accepting state, which the learned local policies cannot sustain. Similarly, in Carlo, demonstrations rarely reach accepting states more than twice. Despite this suboptimality, TiLoIL learns policies that surpass the demonstrations, empirically achieving higher visitation rates of accepting states in both CheetahFlip and Carlo. A key reason for this is that TiLoIL does not attempt to imitate the demonstrator's actions—in fact, we assume that demonstrations consist only of state trajectories (Line 138). This allows TiLoIL to prioritize task satisfaction over action mimicry, leveraging the logical structure of the LTL objective to generalize beyond the demonstrations.
>
> **Q2. If the epsilon-transitions have any meaning in your framework, I'd love to see them incorporated into an example.**
>
> We will revise the paper to include the FlatWorld Stabilization environment as an illustrative example to clarify the role of ε-transitions. In this environment, the LTL task is FG $y$, which requires the agent to eventually reach and then remain in the yellow region. The corresponding LDBA consists of three states. The transition from automaton state 0 to state 1 is an ε-transition, which occurs independently of the MDP state—it represents the logical satisfaction of the "finally" operator in the LTL formula rather than any concrete observation. Once in state 1, the automaton remains there as long as the agent stays in MDP states satisfying $y$ (i.e., remaining in the yellow region). If $y$ becomes false, the automaton transitions to sink state 2. The learned policy must learn to initiate the ε-transition (Def. 2.3) at the right time and then enter the yellow region, taking actions that ensure the agent stays there indefinitely and avoiding the sink state. This illustrates how ε-transitions enable abstract progression in the automaton that is not tied to any specific environment transition, yet must be orchestrated carefully by the policy in concert with the environment dynamics.
>
> **Q3. Can this method be adapted to a setting where the observations of the environment are noisy/partially observed? What if I'm only 90% certain I hit 'blue'?**
>
> Yes! Thanks for the suggestion. Our method can be extended to handle models uncertainty in atomic propositions via belief-based policy learning. In the POMDP setting (environment are noisy/partially observed), the agent has only access an observation distribution $\omega : \mathcal{S} \rightarrow \Delta \mathcal{O}$ induced by the state space $\mathcal{S}$ - the agent observes an observation $o_t \sim \omega(s_t)$ and performs an action based on $o_t$ instead of $s_t$. Denote the agent's observation-action history at time $t$ by $h_t = (o_1, a_1, \ldots, o_{t-1}, a_{t-1}, o_t)$ and the set of all possible histories as $\mathcal{H}$. In principle we can apply any deep RL approach suitable for POMDPs, such as a recurrent policy $a_t \sim \pi(a_t \mid o_t, b_t)$ that conditions on $h_t$ and the current LDBA state $b_t$. However, we don't have access to the ground-truth labeling function $\mathcal{L}$ and therefore cannot accurately evaluate $b_t$. As pointed out by the reviewer, we can assume we access to a model $M : \mathcal{H} \rightarrow \Delta(2^{\mathsf{AP}})$ that predicts probabilities over possible propositional evaluations $\mathcal{L}$ (e.g. 90% certain agent hitting 'blue'). We therefore can use $M$ to predict a distribution over LDBA states $\tilde{b}_t \in \Delta \mathcal{B}$:
> * At $t=0$: we initialize the belief $\tilde{b}_0\$ to be certain of the initial LDBA state $b_0$.
> * For $t > 0$: we propagate belief forward through possible transitions ${P}^\mathcal{B}$, using uncertain information about the environment from $M$:
> > The probability that the LDBA is in state $b$ at time $t$ is computed by summing over all possible prior LDBA states $b'$ and all possible atomic proposition sets $\Sigma \cup \mathcal{E}$, and weighing:
> >
> > * whether $P^\mathcal{B}(b', \sigma) = b\$ (i.e., whether the LDBA would transition from $b'$ to $b$ if evaluation $\sigma$ to atomic propositions were observed),
> > * how likely it was that we were in state $b'$ at $t-1$ (i.e., $\tilde{b}_{t-1}\[b']$),
> > * how likely it is that $\sigma$ is the set of atomic propositions true at time $t$ (from $M(h_t)[\sigma]$).
>
> The decision-making module is a policy $\pi(a_t \mid h_t, \tilde{b}_t)$ that then leverages the inferred belief $\tilde{b}_t$.
>
> At each time step $t$, we can then compute a soft multi-stage reward from the bank of discriminators: $r_t = \sum_{b \in \mathcal{B}} \tilde{b}_t[b] \cdot R((s_t, b))$ for training the agent, where the reward function $R$ is our original multistage reward formulation defined for a product MDP state in Equation (8), Line 302. In this way, $r_t$ smooths out errors in uncertainty in atomic proposition truth values.
>
> To train each stage-specific discriminator $f_b$ (Line 293), we can similarly compute the belief of a trajectory that has passed LDBA states, storing the trajectories in stage buffer with soft weighting  — e.g., weight a sample by $\tilde{b}_t[b]$ when training $f_b$.
>
> We see the integration of our reward shaping into both multi-task frameworks and noisy environments as exciting directions for future work.
>
> **Weakness: The contributions of the paper are not clearly outlined.**
>
> We appreciate this feedback. Our primary contribution is to show that finite-length demonstrations can be used to overcome the exploration bottleneck inherent in learning for LTL objectives over infinite horizons. We will revise the introduction to include an explicit “Contributions” section that makes our key contributions more visible to readers.

---

> > ### Comment · Reviewer_r1Rp · 2025-08-04
> >
> > I thank the authors for their detailed response.
> >
> > I was thinking of a different type of suboptimality - demonstrations that violate the formula by failing to hit some sub-goals for example, but I believe that was covered in answers to the other reviewers. I have no further questions.

---

> ### Author Response · Authors · 2025-08-06
> **Thank you for your feedback!**
>
> We thank the reviewer for the helpful clarification.
>
> **I was thinking of a different type of suboptimality – demonstrations that violate the formula by failing to hit some sub-goals, for example.**
>
> We agree that such demonstrations are particularly challenging. However, our method is designed to degrade gracefully in their presence—suboptimal demonstrations do not collapse the optimization.
>
> As described in Line 5 of Algorithm 1, we still assign demonstration trajectories to LDBA accepting sets \$\mathcal{B}^\star\$, even if they fail to fully complete the task. Demonstrations that fall short of fully satisfying the LTL formula—such as those that progress through some sub-goals or approach accepting states without fully reaching them—can still offer valuable learning signals during training.
>
> To further validate this, we conducted an additional experiment in the *Carlo* environment. This environment involves a self-driving agent trained to follow a circular track counterclockwise while visiting two blue regions repeatedly and avoiding crashes. We trained TiLoIL **using only failed or incomplete demonstrations**, including those that either entered unsafe regions or failed to complete a full loop by missing one blue region. Results (averaged over 5 seeds after 400K training steps) are shown below:
>
> | Environment | TiLoIL (Original Demos) | TiLoIL (Unsafe or Incomplete Demos Only) |
> | ----------- | ------------------- | ------------------------------ |
> | Carlo       | 3.2 ± 1.8           | 2.1 ± 0.5                      |
>
> While there is a performance drop—as expected—TiLoIL remains effective, significantly outperforming the RL for LTL baselines and GAIL, as reported in the main paper. We will include this discussion and the results in the revised version to clarify the method's robustness under such suboptimal demonstrations.
>
> In the revision, as discussed in our initial response, we will add the FlatWorld Stabilization environment as an illustrative example to clarify the role of ε-transitions, discuss how TiLoIL can be adapted to environments with noisy or partial observations, and revise the introduction to include an explicit “Contributions” section. This section highlights our main contribution: demonstrations from suboptimal agents that can only reach LDBA-accepting states a limited number of times (e.g., once or twice in our experiments) can be effectively leveraged to learn policies that satisfy LTL objectives over infinite horizons. We appreciate your comments and suggestions, which strengthen the clarity and scope of our work!

---

### Note · Authors · 2025-08-12

We thank all reviewers and the AC for their constructive feedback and engagement. The main concerns raised involve (1) clarifying TiLoIL's assumptions and contributions, and (2) demonstrating scalability.

We contribute a novel imitation learning algorithm leveraging demonstrations from suboptimal agents that can only reach LDBA-accepting states a limited number of times (e.g., once or twice) to learn policies that visit accepting states infinitely often, thereby satisfying LTL objectives over infinite horizons. The generalization from such sparse, finite, suboptimal behaviors to policies that achieve LTL satisfaction is important in tasks with complex logical structure [Reviewer ifPj]. The reviewers raised concerns about cases where demonstrations fail to reach any accepting state or cover varying accepting conditions. We clarified that TiLoIL can leverage the agent's own exploratory behaviors—those progressing beyond certain LDBA states—as positive trajectories to train discriminators that guide learning for trajectories that fail to meet such states. We demonstrated this in the Carlo task, where demonstrations were unsafe or incomplete and failed to reach the accepting state, yet TiLoIL remained effective, outperforming the baselines [Reviewer r1Rp]. However, in such scenarios, the exploration capability of the underlying RL algorithm can become a bottleneck. We further investigated it through ablation studies on the additional Symbol-like and Cookie-like environments [Reviewer Tszg] and will clarify this limitation in the revision. Additionally, we showed that integrating TiLoIL with potential-based reward shaping achieves state-of-the-art results without any demonstrations, highlighting a promising future direction [Reviewer uyFt].

Reviewer uyFt raised concerns about scalability. We addressed this in two ways: (1) introducing a shared discriminator across LDBA states, reducing complexity compared to one discriminator per state, with experimental results demonstrating its feasibility; and (2) evaluating TiLoIL on more complex LTL specifications with larger LDBAs (e.g., 16Rooms, FlatWorld Disjunction, Symbol-like, and Cookie-like), where TiLoIL was the only method among all baselines to meaningfully solve the tasks. The results on the Symbol-like and Cookie-like tasks also demonstrate TiLoIL's ability to scale to settings with dynamic placement of regions or objects.

We will revise the paper according to the revision plans provided to each reviewer.

---

### Decision · Program_Chairs · 2025-09-17

**Decision:**

Accept (poster)

**Comment:**

This paper proposes TiLoIL, an imitation learning method designed to handle infinite-horizon tasks specified using Linear Temporal Logic (LTL). The approach leverages segmented demonstrations and product MDP structure to generate dense rewards and guide exploration. Trajectories are segmented at accepting states, and stages within those segments are used to refine the discriminator training in a GAIL-style framework. Experiments across multiple domains, including continuous control and gridworld settings, show faster convergence and improved performance compared to baselines without LTL guidance.

**Strengths:** The paper addresses an important and timely problem at the intersection of reinforcement learning and formal task specification. It clearly motivates the difficulty of combining LTL with finite demonstrations and proposes a principled way to densify sparse rewards through segmentation and staging. The method is well described, and the ablation studies highlight the value of the segmentation mechanism, particularly in cyclical tasks. Related work is carefully discussed, and the paper demonstrates solid scholarship. During the rebuttal, the authors provided thorough clarifications, new experiments, and detailed responses that improved the clarity of contributions and addressed most reviewers' questions.

**Weaknesses:** While the contributions are meaningful, the evaluation remains somewhat limited in scope. The tested environments are fixed, which simplifies the benefit of demonstrations, and the method’s scalability to more complex LTL specifications with larger automata is not fully established. The approach assumes access to near-optimal demonstrations, and it is unclear how robust it would be to sub-optimal or incomplete ones. While the authors suggest that stronger exploratory RL algorithms could compensate for this, such an argument does not fully address the concern.

**Final Recommendation:** Overall, the paper makes a novel and interesting contribution toward learning policies from demonstrations under infinite-horizon LTL constraints. The authors engaged very constructively during the rebuttal, providing additional results and clarifications. I recommend acceptance (poster). The authors should, however, clearly state in the final version the limitations of their evaluation (fixed environments) and discuss possible directions for overcoming this, as well as strengthen the discussion of assumptions on demonstrations and related work.